# Topological heavy fermions in magnetic field

Keshav Singh [1,2], Aaron Chew[3], Jonah Herzog-Arbeitman [3], B. Andrei Bernevig [3,4,5] & Oskar Vafek [1,2] ✉

The recently introduced topological heavy fermion model (THFM) provides a means for interpreting the low-energy electronic degrees of freedom of the magic angle twisted bilayer graphene as hybridization amidst highly dispersing topological conduction and weakly dispersing localized heavy fermions. In order to understand the Landau quantization of the ensuing electronic spectrum, a generalization of THFM to include the magnetic field $B$ is desired, but currently missing. Here we provide a systematic derivation of the THFM in $B$ and solve the resulting model to obtain the interacting Hofstadter spectra for single particle charged excitations. While naive minimal substitution within THFM fails to correctly account for the total number of magnetic subbands within the narrow band i.e., its total Chern number, our method—based on projecting the light and heavy fermions onto the irreducible representations of the magnetic translation group— reproduces the correct total Chern number. Analytical results presented here offer an intuitive understanding of the nature of the (strongly interacting) Hofstadter bands.

Since the discovery of the remarkable phase diagram of the magic angle twisted bilayer graphene (MATBG)[1,2], substantial effort[3–32] has been devoted to understanding its rich physics. The presence of topological narrow bands within this system[23,33–35] provides a novel platform to study the interplay between strong electron correlations and band topology. The recently introduced topological heavy fermion model (THFM) for MATBG[36,37] bridges the contrary signatures of localized[38,39] and delocalized physics[40,41] reported via STM and transport measurements[42,43]. Within THFM the low energy electrons are viewed as a result of the hybridization between heavy $p_x \pm i p_y$-like Wannier states, localized at the AA stacking sites, and topological conduction fermions, denoted by $f$ and $c$, respectively, in analogy to heavy fermion materials[36]. Among its other features, THFM allows for an intuitive explanation of the charged excitation spectra[36] at integer fillings hitherto obtained via strong coupling expansion of projected models[18,25].

The large moiré period of ~13 nm in MATBG has revealed a sequence of broken symmetry Chern insulators yielding a plethora of finite magnetic field (**B**) induced phases at lower fluxes[42,44–51] and has showcased, for the first time, reentrant correlated Hofstadter states at magnetic fields as low as 31T[52]. Thus it becomes important to better

understand the interplay of correlations and band topology in the presence of a perpendicular **B** field. Theoretical studies have previously focused on non-interacting[53–55] and strong coupling[56–58] regimes. Although exact, each employed considerable numerical analysis, restricting a deeper physical understanding of the mechanism for Landau quantization.

In this paper, we show how one can understand the Landau quantization of the strong coupling spectra in terms of hybridization amidst Landau levels (LLs) of $c$ fermions and hybrid Wannier states of $f$ fermions. We find that only a particular number of $f$ fermion momentum channels are allowed to hybridize to $c$ fermion LLs, with coupling strength which decreases with increasing **B** and increasing LL index $m$. Moreover, through our analysis we can clearly understand the reason why a naive minimal coupling is unable to recover the correct total Chern number of the narrow bands. In the flat band limit of THFM, our framework allows for an exact solution including the dominant interactions and analytically explains the total Chern number. Even for cases with a non-trivial Chern number, we explicitly demonstrate the dependence of total number of states on the magnetic field as is expected by the Streda formula[59]. Although going away from the flat band limit requires numerics, given the simple structure

[1]National High Magnetic Field Laboratory, Tallahassee, FL 32310, USA. [2]Department of Physics, Florida State University, Tallahassee, FL 32306, USA. [3]Department of Physics, Princeton University, Princeton, NJ 08544, USA. [4]Donostia International Physics Center, P. Manuel de Lardizabal 4, 20018 Donostia-San Sebastian, Spain. [5]IKERBASQUE, Basque Foundation for Science, Bilbao, Spain. ✉e-mail: vafek@magnet.fsu.edu

of our Hamiltonian, we are still able to compute the spectrum to unprecedentedly small fluxes and find it to be well captured by the analytical solution in the flat band limit, $M = 0$, which can be taken all the way to $\mathbf{B} = 0$ as shown for narrow band filling factors $\nu = 0$, $-1$, and $-2$ in this paper.

The formulas we derive are general for any rational value of $\phi/\phi_0$, with $\phi$ being the flux through the unit cell and $\phi_0$ being the flux quantum $hc/e$, but we focus our analysis on the $1/q$ flux sequence and low $\mathbf{B}$ where the results become particularly transparent. Our analysis as well unveils the physical nature of the anomalous low energy mode which is seen to be almost $\mathbf{B}$-independent, also observed in previous numerics[56], as the anomalous zero-LL of a massless Dirac particle, a key ingredient of the topological heavy fermion picture of MATBG. Although this work deals directly with THFM, our methods apply more generally.

## Results

### Review of THFM and the key challenge

The THFM in momentum space is given by[36]

$$
\hat{H}_0 = \sum_{|\mathbf{k}|<\Lambda_c} \sum_{\tau s} \sum_{aa'=1}^{4} H_{aa'}^{c,\tau}(\mathbf{k}) \tilde{c}_{\mathbf{k}a\tau s}^{\dagger} \tilde{c}_{\mathbf{k}a'\tau s}
$$
$$
+ \sum_{|\mathbf{k}|<\Lambda_c} \sum_{\tau s} \sum_{a=1}^{4} \sum_{b=1}^{2} \left( e^{-\frac{1}{2}\mathbf{k}^2\lambda^2} H_{ab}^{cf,\tau}(\mathbf{k}) \tilde{c}_{\mathbf{k}a\tau s}^{\dagger} \tilde{f}_{\mathbf{k}b\tau s} + \text{h.c.} \right). \quad (1)
$$

Here $\Lambda_c$ is the momentum cutoff for $c$ fermions while $f$ fermions, whose bandwidth is negligibly small, reside in the entire moiré Brillouin zone (mBZ). The tilde on the fermionic creation and annihilation operators indicates that they are at $\mathbf{B} = 0$. The parameter $\lambda \approx 0.38 L_m$ is a damping factor proportional to the spatial extent of the localized Wannier functions[36], with $L_m$ being the moiré period; $\tau = +1(-1)$ represents graphene valley $\mathbf{K}(\mathbf{K}')$ and $s$ spin $\uparrow$, $\downarrow$. The $c$–$c$ and $c$–$f$ couplings are

$$
H^{c,1} = \begin{pmatrix} 0 & 0 & v_* k & 0 \\ 0 & 0 & 0 & v_* \bar{k} \\ v_* \bar{k} & 0 & 0 & M \\ 0 & v_* k & M & 0 \end{pmatrix}, \; H^{cf,1} = \begin{pmatrix} \gamma & v_*' \bar{k} \\ v_*' k & \gamma \\ 0 & 0 \\ 0 & 0 \end{pmatrix}, \quad (2)
$$

where $k = k_x + ik_y$ and $\bar{k} = k_x - ik_y$. The explicit $\mathbf{k}$ dependence in $H^{c,1}(\mathbf{k})$ and $H^{cf,1}(\mathbf{k})$ above has been suppressed for brevity. The parameters $v_*$, $v_*'$, $M$ and $\gamma$ were determined from the Bistritzer-MacDonald[60] (BM) model in ref. 36. The bandwidth of narrow bands is set by $2|M|$ and the gap between the narrow bands and the remote bands is $|\gamma| - |M|$. The couplings at the opposite graphene valley (i.e., at $\tau = -1$) can be obtained by replacing $k \leftrightarrow -\bar{k}$ in Eq. (2).

In order to illustrate the key challenge to promoting the model to non-zero $B$, we consider a simplified case wherein we set the bandwidth and the spatial extent of the localized Wannier functions to zero, i.e., $M = \lambda = 0$ in Eqs. (1) and (2). As argued below and as shown in the Supplementary Note 8, the conclusions reached hold even for a general case without making this simplification. Following a naive minimal substitution, we promote $k_x + ik_y \rightarrow -i\sqrt{2}\hat{a}/\ell$[61], where the magnetic length is $\ell = \sqrt{\hbar c/(eB)}$ and $\hat{a}$ is the Landau level (LL) lowering operator. The eigenstates of thus minimally substituted Hamiltonian $\begin{pmatrix} H^{c,1} & H^{cf,1} \\ H^{cf,1\dagger} & 0 \end{pmatrix}$ can be obtained exactly. It can be readily verified that the zero modes take the form $(a_{1,m}|m\rangle, a_{2,m}|m-1\rangle, a_{3,m}|m+1\rangle, a_{4,m}|m-2\rangle, .. b_{1,m}|m\rangle, b_{2,m}|m-1\rangle)^T$, where $|n\rangle$ denotes $n$th-LL and $a_{\alpha,m}$ and $b_{\beta,m}$ are complex coefficients with $\alpha \in \{1, ..., 4\}$ and $\beta \in \{1, 2\}$. The LL index $m \in \{0, ..., m_*\}$ where $m_*$ denotes its upper cut-off. For $m = 0$, the non-zero coefficients are $a_{3,0} = -i\gamma\ell$ and $b_{1,0} = \sqrt{2}v_*$, while for $m = 1$ they are $a_{3,1} = 2v_*'^2 - \gamma^2\ell^2$, $b_{1,1} = -2i\gamma\ell v_*$ and $b_{2,1} = 2\sqrt{2}v_* v_*'$. For each

$m \geq 2$, there are two zero modes whose non-zero coefficients are $a_{3,m} = \sqrt{2}\sqrt{m(m-1)}v_*'$, $a_{4,m} = i\gamma\ell\sqrt{m+1}$, $b_{2,m} = \sqrt{2}\sqrt{m^2-1}v_*$ and $a_{3,m} = -i\gamma\ell\sqrt{m-1}$, $a_{4,m} = \sqrt{2}\sqrt{m(m+1)}v_*'$, $b_{1,m} = \sqrt{2}\sqrt{m^2-1}v_*$, respectively. Including the anomalous zero-LL of the $c$–$c$ coupling, $(0,0,|0\rangle,0,0,0)^T$, we have a total of $2m_*+1$ zero modes within the narrow bands.

These zero modes are well separated from the remote subbands by a gap that limits to $|\gamma|$ as $\mathbf{B} \rightarrow 0$. This gap cannot close in the stated limit even if we relax the above mentioned assumptions. The value of $m_*$ is typically determined by requiring that the LL spectrum converges in the energy range of interest. For us this energy window includes the narrow bands and perhaps several LLs from the remote bands. However, increasing $m_*$ results in an unbounded increase in the number of LLs within the narrow band energy range as seen from our exact result. In other words, since each LL contains $\phi/\phi_0$ states per moiré unit cell, the zero modes would accommodate $(2m_* + 1)\phi/\phi_0$ states per moiré unit cell for each spin and valley. But the total Chern number of the narrow bands at $\mathbf{B} = 0$ vanishes which means that the zero modes should accommodate precisely two states per moiré unit cell for each spin and valley *independent* of $\mathbf{B}$[59]. This demonstrates that the naive minimal coupling is unable to account for the correct number of magnetic subbands within the narrow bands for an arbitrary $m_*$.

In the next sections, we introduce the framework for systematically promoting THFM to non-zero $\mathbf{B}$ and naturally solve the problem illustrated above. Our approach also provides a deeper understanding of the nature of the Hofstadter subbands. This framework is also extended to include interactions at a mean-field (MF) level. We do so by illustrating the Landau quantization for the "one-shot" Hartree-Fock (HF) bands obtained using the MF Hamiltonian for the parent valley polarized (VP) state[36] at three different integer fillings of the narrow bands.

### Basis states at non-zero magnetic field

As illustrated in the previous section, the naive minimal coupling is inadequate. In order to develop a systematic framework for THFM at non-zero $B$, we begin by carefully constructing the basis states in the way that takes into account the nature of the $c$ and $f$ fermions. In addition, our construction takes advantage of the magnetic translation symmetry of the underlying Hamiltonian[60]. This not only helps us to label our states using good quantum numbers but also allows us to transparently keep track of the total number of states at finite $\mathbf{B}$.

We start by briefly reviewing the magnetic translation symmetry. In the presence of an out-of-plane magnetic field, employed via Landau gauge vector potential $\mathbf{A} = (0, Bx, 0)$, the minimally coupled BM Hamiltonian[60], $H_{BM}^\tau(\mathbf{p} - \frac{e}{c}\mathbf{A})$, preserves the translational symmetry with respect to the primitive moiré lattice vector $\mathbf{L}_2 = L_m(0, 1)$ but translation with respect to the primitive moiré lattice vector $\mathbf{L}_1 = L_m(\frac{\sqrt{3}}{2}, \frac{1}{2})$ needs to be accompanied by a gauge transformation. In other words, if $f(\mathbf{r})$ is an eigenstate of $H_{BM}^\tau(\mathbf{p} - \frac{e}{c}\mathbf{A})$, then so is $\hat{t}_{\mathbf{L}_2} f(\mathbf{r}) = f(\mathbf{r} - \mathbf{L}_2)$ and $\hat{t}_{\mathbf{L}_1} f(\mathbf{r}) = \exp\left(i\frac{L_{1x}y}{\ell^2}\right) f(\mathbf{r} - \mathbf{L}_1)$ with the same eigenvalue (also see supplementary note 1 for details). These operators do not commute as $\hat{t}_{\pm \mathbf{L}_2}\hat{t}_{\mathbf{L}_1} = \exp\left(\mp 2\pi i\frac{\phi}{\phi_0}\right)\hat{t}_{\mathbf{L}_1}\hat{t}_{\pm\mathbf{L}_2}$, where the flux through the moiré unit cell is $\phi = BL_{1x}L_m$. However, for $\phi/\phi_0 = p/q$, with $p$ and $q$ being relatively prime integers, we have the commuting set of magnetic translation(MT) operators $[\hat{t}_{\mathbf{L}_2}^q, \hat{t}_{\mathbf{L}_1}] = 0$, which we use to label our basis states.

We can now proceed to construct the non-zero $\mathbf{B}$ basis for $f$ fermions by utilizing MT. At $\mathbf{B} = 0$, the basis for $f$s is composed of two Wannier functions $W_{\mathbf{R},b\tau}(\mathbf{r}) = W_{\mathbf{0},b\tau}(\mathbf{r} - \mathbf{R})$ in each moiré unit cell which behave as $p_x \pm ip_y$ orbitals sitting on the AA stacking sites spanned by moiré triangular lattice vector $\mathbf{R}$. The highly localized nature of these states and the trivial topology of their bands allow us to construct a complete basis for the $f$s at $\mathbf{B} \neq 0$ using the hybrid Wannier method[56,62]. To this end we first construct hybrid Wannier states (hWs) out of

$W_{\mathbf{0},b\tau}(\mathbf{r})$ by a repeated application of the translation operator $\hat{t}_{\mathbf{L}_2}$ (as seen in Eq. (3)). The hWs are Bloch extended in the $y$-direction, i.e., along $\mathbf{L}_2$, while localized in the $x$-direction with the localization center at the origin. Note that $\mathbf{A}$ respects the translational symmetry along $y$. Moreover, near the origin where $\mathbf{A}$ is small at small $B$, hWs must have a large overlap with the $\mathbf{B} \neq 0$ magnetic subbands that emanate out of $\mathbf{B} = 0$ bands of $f$s i.e., the $\mathbf{B} \neq 0$ Hilbert space of $f$'s that we pursue[62]. The rest of the basis is then generated similarly by projecting the hWs onto irreducible representations (irreps) of the magnetic translation group (MTG) as

$$\eta_{b\tau k_1 k_2}(\mathbf{r}) = \frac{1}{\sqrt{\mathcal{N}}} \sum_{s,n \in \mathbb{Z}} e^{2\pi i(sk_1 + nk_2)} \hat{t}_{\mathbf{L}_1}^s \hat{t}_{\mathbf{L}_2}^n W_{\mathbf{0},b\tau}(\mathbf{r}). \tag{3}$$

Clearly, $\eta_{b\tau k_1 k_2}$ is a simultaneous eigenstate of $\hat{t}_{\mathbf{L}_1}$ and $\hat{t}_{\mathbf{L}_2}^q$ with eigenvalues $e^{-2\pi i k_1}$ and $e^{-2\pi i q k_2}$, respectively. Thus, $k_{1,2}$ labels the momentum along the primitive reciprocal lattice vectors $\mathbf{g}_{1,2}$, where $\mathbf{g}_1 = \frac{4\pi}{\sqrt{3}L_m}(1,0)$ and $\mathbf{g}_2 = \frac{2\pi}{\sqrt{3}L_m}(-1,\sqrt{3})$. For the $f$s $k_{1,2} \in [0,1)$ i.e., there are two $f$s per moire unit cell; the normalization factor $\mathcal{N} = s_{tot} n_{tot}$, where $s_{tot}$ and $n_{tot}$ denote the total count of $s$ and $n$ (see Supplementary Note 3 for details). The states with different $k_1$ and $[k_2]_{1/q}$ are guaranteed to be orthogonal which is apparent through their eigenvalues under the MTs, where $[b]_a$ denotes $b$ modulo $a$. In Supplementary Note 3, we prove that the overlap between states with $k_2$ differing by integral multiples of $1/q$ is very small, i.e., to an excellent approximation, these states are also orthogonal. This stems from the extremely well-localized nature of the $\mathbf{B} = 0$ Wannier states.

In order to construct the non-zero $\mathbf{B}$ basis for $c$ fermions, we recall that the $\mathbf{B} = 0$ basis for the $c$s is composed of four $\mathbf{k} \cdot \mathbf{p}$ Bloch states $e^{i\mathbf{k} \cdot \mathbf{r}} \tilde{\Psi}_{\Gamma a\tau}(\mathbf{r})$, where $\tilde{\Psi}_{\Gamma a\tau}$ is the Bloch state at the $\Gamma$ point in mBZ[36]. We can extend the $c$s to non-zero $B$ by multiplying $\tilde{\Psi}_{\Gamma a\tau}$ by LL wavefunctions, a result obtained when the $\mathbf{k} \cdot \mathbf{p}$ method is extended to $\mathbf{B} \neq 0$[63]. So as to use the same quantum numbers as for $f$s, we also project the $c$'s LL wavefunctions $\Phi_m$ onto the (orthonormal) irreps of the MTG as

$$\chi_{k_1 k_2 m}(\mathbf{r}) = \frac{1}{\sqrt{\ell L_m}} \frac{1}{\sqrt{\mathcal{N}}} \sum_{s \in \mathbb{Z}} e^{2\pi i s k_1} \hat{t}_{\mathbf{L}_1}^s \Phi_m(\mathbf{r}, k_2 \mathbf{g}_2). \tag{4}$$

Again, $\chi_{k_1 k_2 m}$ is a simultaneous eigenstate of $\hat{t}_{\mathbf{L}_1}$ and $\hat{t}_{\mathbf{L}_2}^q$ with eigenvalues $e^{-2\pi i k_1}$ and $e^{-2\pi i q k_2}$, respectively. Here $k_1 \in [0,1)$, but unlike in Eq. (3), $k_2 \in [0, \frac{p}{q})$ i.e., there are $\phi/\phi_0 = p/q$ states per moiré unit cell in each Landau level (see Supplementary Note 3 for details on orthonormality and the domain of $k_{1(2)}$). $\Phi_j(\mathbf{r}, k_2 \mathbf{g}_2) = e^{2\pi i k_2 \frac{y}{L_m}} \varphi_j(x - k_2 \frac{2\pi \ell^2}{L_m})$, the harmonic oscillator (h.o.) wavefunctions $\varphi_m(x) = e^{-x^2/2\ell^2} H_m(x/\ell)/\pi^{\frac{1}{4}} \sqrt{2^m m!}$ with Hermite polynomials $H_m$. The $k_2$ induced offset in the h.o. wavefunctions is $2\pi k_2 \ell^2/L_m = q k_2 L_{1x}/p$. Although the LL wavefunction $\Phi_m$ is an eigenstate of $\hat{t}_{\pm \mathbf{L}_2}$, the function $\chi_{k_1 k_2 m}$ is not. Instead, $\hat{t}_{\pm \mathbf{L}_2} \chi_{k_1 k_2 m}(\mathbf{r}) = e^{\mp 2\pi i k_2} \chi_{[k_1 \mp \frac{p}{q}] k_2 m}(\mathbf{r})$. We utilize this identity in the proceeding sections (see Supplementary Note 4B, Eqs. (92)–(94) for derivation).

Since the $q\mathbf{L}_2$ translations break up the $k_2$ domain into units of width $\frac{1}{q}$, from here on we use the label $k = (k_1, k_2)$ with $k_1 \in [0,1)$ and we fix $k_2 \in [0, \frac{1}{q})$. The original $k_2$ domains are then accessed using labels $r' \in \{0, \dots q-1\}$ and $r \in \{0, \dots, p-1\}$, denoting the magnetic strip $[\frac{r'}{q}, \frac{r'+1}{q})$ and $[\frac{r}{q}, \frac{r+1}{q})$ along $\mathbf{g}_2$ for $\eta$ and $\chi$, respectively. We thus relabel the states as $\eta_{b\tau k r'}(\mathbf{r})$ and $\chi_{krm}(\mathbf{r})$, respectively. Having assembled the low energy basis at $\mathbf{B} \neq 0$, we now expand the low energy field at a given spin $s$ and valley $\tau$ as

$$\psi_{\tau,s}(\mathbf{r}) = \sum_{k \in [0,1) \otimes [0, \frac{1}{q})} \left( \sum_{b=1}^{2} \sum_{r'=0}^{q-1} \eta_{b\tau k r'}(\mathbf{r}) f_{b\tau k r' s} \right.$$
$$\left. + \sum_{a=1}^{4} \sum_{m=0}^{m_{a,\tau}} \sum_{r=0}^{p-1} \Psi_{a\tau}(\mathbf{r}) \chi_{krm}(\mathbf{r}) c_{a\tau krms} \right), \tag{5}$$

where $\Psi_{a\tau}(\mathbf{r}) = \sqrt{A_{tot}} \tilde{\Psi}_{\Gamma a\tau}(\mathbf{r})$ with $A_{tot}$ being the total sample area, and $f_{b\tau k r' s}$ and $c_{a\tau krms}$ denote the annihilation operators for the $\mathbf{B} \neq 0$ $f$ and $c$ basis states, respectively. Anticipating the appearance of anomalous Dirac LLs for the topological $c$ fermions, we allow for the $a$ dependence of the upper cutoff on the LL index at each valley $\tau$, denoted by $m_{a,\tau}$ above, with $m_{1,+1} = m_{2,-1} = m_\star + 1$, $m_{2,+1} = m_{1,-1} = m_\star$, $m_{3,+1} = m_{4,-1} = m_\star + 2$ and $m_{4,+1} = m_{3,-1} = m_\star - 1$. As discussed in Supplementary Note 3, the choice of $m_\star$, although arbitrary, needs to be below an upper-bound to ensure the orthogonality amidst the $c$-states $\Psi_{a\tau}(\mathbf{r})\chi_{krm}(\mathbf{r})$. This is because it relies on the fact that their overlaps are exponentially small in $\ell^2 \mathbf{g}^2$ as long as the LL index $m$ is held below an upper cutoff $m_\star \lesssim q/2$, where $\mathbf{g}$ is the reciprocal moiré lattice vector.

## Non-interacting Hamiltonian at $\mathbf{B} \neq 0$

The single particle THFM at $\mathbf{B} \neq 0$ can be computed using the low energy fields derived in the previous section (Eq. (5))

$$\hat{H}_0^B = \sum_{\tau,s} \int d^2 \mathbf{r} \psi_{\tau,s}^\dagger(\mathbf{r}) H_{BM}^\tau \left( \mathbf{p} - \frac{e}{c} \mathbf{A} \right) \psi_{\tau,s}(\mathbf{r})$$
$$\approx \sum_{\tau,s} H_{cc}^{\tau,s} + \sum_{\tau,s} \left( H_{cf}^{\tau,s} + \text{h.c.} \right), \tag{6}$$

where the $f$–$f$ coupling is neglected in the last line because it is negligibly small (this is also the case at $\mathbf{B} = 0$ in Eq. (1)). The $c$–$c$ and $c$–$f$ couplings are

$$H_{cc}^{\tau,s} = \sum_{k \in [0,1) \otimes [0,\frac{1}{q})} \sum_{a,a'=1}^{4} \sum_{m=0}^{m_{a,\tau}} \sum_{m'=0}^{m_{a',\tau}} \sum_{r,\bar{r}=0}^{p-1} \tilde{h}_{[amr],[a'm'\bar{r}]}^\tau(k) c_{a\tau krms}^\dagger c_{a'\tau k\bar{r}m's}, \tag{7}$$

$$H_{cf}^{\tau,s} = \sum_{k \in [0,1) \otimes [0,\frac{1}{q})} \sum_{a=1}^{4} \sum_{b=1}^{2} \sum_{m=0}^{m_{a,\tau}} \sum_{r=0}^{p-1} \sum_{r'=0}^{q-1} h_{[amr],[br']}^\tau(k) c_{a\tau krms}^\dagger f_{b\tau kr's}. \tag{8}$$

The matrix element for $c$–$c$ coupling $\tilde{h}_{[amr],[a'm'\bar{r}]}^\tau(k) = \langle \Psi_{a\tau}\chi_{krm} | H_{BM}^\tau(\mathbf{p} - \frac{e}{c}\mathbf{A}) | \Psi_{a'\tau}\chi_{k\bar{r}m'} \rangle$ takes the same form as obtained by the direct minimal substitution in $c$–$c$ coupling in Eq. (2) and expanding in LL basis, as is expected from $\mathbf{k} \cdot \mathbf{p}$[63]:

$$\tilde{h}_{[amr],[a'm'\bar{r}]}^\tau(k) = \delta_{r\bar{r}} \begin{pmatrix} 0_{2\times2} & h_{mm'}^{\tau,c} \\ \sigma_x h_{mm'}^{\tau,c} \sigma_x & M\delta_{mm'}\sigma_x \end{pmatrix}_{aa'}, \tag{9}$$

where the Pauli matrix $\sigma_x$ acts on the $c$ orbitals and

$$h_{mm'}^{+1,c} = i\frac{\sqrt{2}v_\star}{\ell} \begin{pmatrix} -\sqrt{m'}\delta_{m+1,m'} & 0 \\ 0 & \sqrt{m}\delta_{m,m'+1} \end{pmatrix} \tag{10}$$

with $h_{mm'}^{-1,c} = -\sigma_x h_{mm'}^{+1,c}\sigma_x$. For $M = 0$, we recover the LLs of two massless Dirac particles, with two zero LLs at each valley (see Supplementary Note 4A for details of derivation).

As discussed in the previous section, there are two $f$-states per moiré unit cell per valley for each spin projection. On the other hand, for each $c$-LL there are $p/q$ states per moiré unit cell per valley for each spin projection. In order to understand the hybridization between these states that, together with the $c$–$c$ coupling, gives rise to an isolated band of states whose total number is independent of $\mathbf{B}$—because its total Chern number vanishes[59]—we need to carefully analyze the $c$–$f$ coupling. Although formidable at first sight, it is actually possible to find an analytical expression for this matrix element $h_{[amr][br']}^\tau(k) = \langle \Psi_{a\tau}\chi_{krm} | H_{BM}^\tau(\mathbf{p} - \frac{e}{c}\mathbf{A}) | \eta_{b\tau kr'} \rangle$ and thus determine the $c$–$f$ coupling at non-zero $\mathbf{B}$. The "Evaluation of the $c$–$f$ matrix elements at $\mathbf{B} \neq 0$" section provides the key steps for the derivation which we omit here for brevity. The result can be cast in a closed form expression

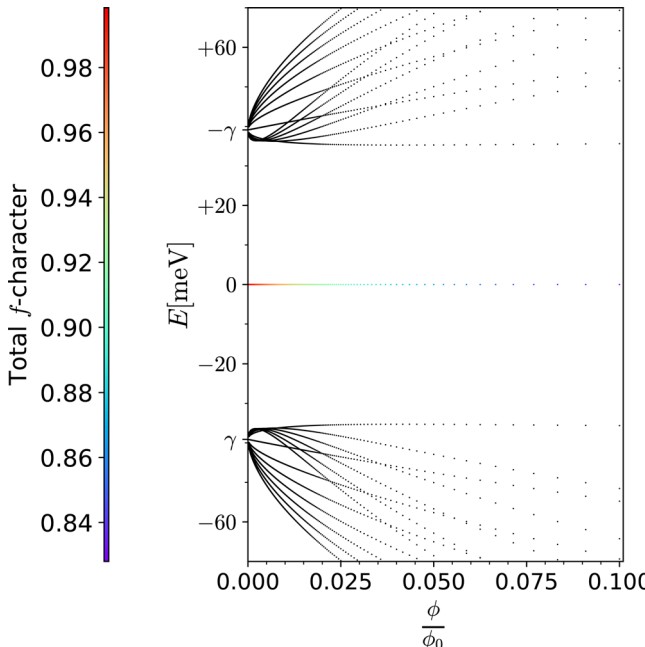

**Fig. 1 | Non-interacting flat band Hofstadter spectrum.** The spin-valley degenerate non-interacting Hofstadter spectrum for THFM at $w_0/w_1 = 0.7$ in the flat band limit $M = 0$. For illustration, we have fixed $m_* = 5$ so that the $\mathbf{B} \to 0$ energies for remote magnetic subbands, i.e., $\pm\gamma$, are tractable. The value of parameters used are $\gamma = -39.11$ meV, $v'_* = 1.624$ eV · Å, $v_* = -4.483$ eV.Å and $\lambda = 0.3792 L_m$. Total $f$-character color labeling on the left, unlike in the rest of the figures, represents the total $f$ weight of the flat bands composed of zero modes. We sum over the $f$-weights of each zero mode and normalize it by the total number of zero modes, i.e., $2q$. For the coupled modes obtained using ansätze in Eqs. (20)–(22), the $f$-weight is obtained as $|c_5^{(\mu)}|^2 + |c_6^{(\mu)}|^2$, after normalizing the eigenvector. The $f$-weight for the decoupled $f$-modes is 1, while that of the anomalous $c$ in Eq. (19) is zero. The remote bands in black do not correspond to the above color labeling. The $y$-label $E$[meV] represents energy of subbands in meV.

which for $p = 1$ and for $\ell \gg \lambda$ reads

$$h_{[am0],[br']}^{\tau}(k) = \begin{pmatrix} \gamma \Upsilon_{m,r'}(k)\sigma_0 + h_{m,r'}^{\tau,cf}(k) & \\ & 0_{2\times 2} \end{pmatrix}_{ab}, \quad (11)$$

with $h_{m,r'}^{+1,cf}(k) =$

$$i\frac{\sqrt{2}v'_*}{\ell}\begin{pmatrix} 0 & \sqrt{m}\Upsilon_{m-1,r'}(k) \\ -\sqrt{m+1}\Upsilon_{m+1,r'}(k) & 0 \end{pmatrix}, \quad (12)$$

and $h_{m,r'}^{-1,cf}(k) = -\sigma_x h_{m,r'}^{+1,cf}(k)\sigma_x$, where $\sigma_x$ acts in orbital space of $c$ and $f$ fermions. The matrix $\Upsilon_{m,r'}(k)$ is given as

$$\Upsilon_{m,r'}(k) = \sqrt{\frac{L_{1x}}{\ell}}e^{i\pi r'_q(k_2 - 2k_1)}e^{i\pi r'_q(r'_q - 1)\frac{1}{2q}}$$

$$e^{-2\pi^2\frac{\lambda^2}{L_m^2}\left(k_2 + \frac{r'_q}{q}\right)^2}\mathcal{F}_m\left(\lambda, (r'_q + qk_2)L_{1x}\right), \quad (13)$$

where $r'_q = \mathrm{sgn}_+\left(\frac{q}{2} - r'\right)\min[r', q - r']$ with $\mathrm{sgn}_+(x)$ being the usual sign function except at 0 where it evaluates to 1, and

$$\mathcal{F}_m(\lambda, x_0) = \frac{1}{\pi^{\frac{1}{4}}\sqrt{2^m m!}}\sqrt{\frac{\ell^2}{\ell^2 + \lambda^2}}e^{-\frac{x_0^2}{2(\ell^2 + \lambda^2)}}$$

$$\times \mathcal{H}_m\left(\frac{-2x_0\ell}{\ell^2 + \lambda^2}, -1 + \frac{2\lambda^2}{\ell^2 + \lambda^2}\right). \quad (14)$$

The two variable Hermite polynomials[64] are given by $\mathcal{H}_m(x,y) = m!\sum_{k=0}^{\lfloor\frac{m}{2}\rfloor}(x^{m-2k}y^k)/((m-2k)!k!)$, where $\lfloor m \rfloor$ denotes the floor function

at $m$. Their relation to the Hermite polynomials used above is $H_m(x) = \mathcal{H}_m(2x, -1)$.

Although we can significantly simplify the form of $\hat{H}_0^B$ and gain a deeper analytical understanding of our solution as we do in next section, one can already use the above expressions to obtain the Hofstadter spectrum for THFM numerically. Such numerical calculation recovers the correct total number of states within the narrow band energy window, i.e., 2 per moiré unit cell per valley for each spin projection regardless of the value of $m_*$, thus solving the key challenge outlined earlier. As illustrated in Fig. 1 for $M = 0$, these zero modes are well separated from the remote bands by a gap that limits to $|\gamma|$ as $\mathbf{B} \to 0$. The results are qualitatively the same for $M \neq 0$ as shown in Supplementary Fig. 10b except the zero modes split into a group of states with a width set by $2|M|$ as expected. In the following sections, we elucidate the nature of the Hofstadter subbands by carefully casting the $\mathbf{B} \neq 0$ Hamiltonian in terms of coupled and decoupled modes of $f$ fermions. This not only helps us to obtain an exact solution in the flat band limit but also to understand the total Chern number via straightforward analytical arguments.

**Analytical results for the non-interacting Hamiltonian at B ≠ 0**
As mentioned, for $M = 0$ we find two isolated zero modes per moiré unit cell per valley for each spin projection from numerical calculation. In order to obtain these zero modes analytically, we start by noting that the general form of our Hamiltonian at $\mathbf{B} \neq 0$ presented in the previous section immediately implies a certain lower bound on their number. Within each valley and for each spin projection, the Hamiltonian matrix at a given $k$ has the form $\begin{pmatrix} C & F \\ F^\dagger & 0 \end{pmatrix}$ where $C$ is a square matrix of dimension $4m_* + 6$ and $F$ is a $(4m_* + 6) \times 2q$ rectangular matrix; the last $2q \times 2q$ block is filled with zeros. This automatically guarantees a lower bound on the number of zero modes equal to the difference in the number of $F$'s columns and rows, as is easily established by considering the singular value decomposition (SVD) of $F$ (see, e.g., ref. 65). Moreover, as seen in the Eq. (11), $F$ has the form $\begin{pmatrix} F' \\ 0_{(2m_* + 3)\times 2q} \end{pmatrix}$ where $F'$ is a $(2m_* + 3) \times 2q$ rectangular matrix. Therefore, half of the singular values of $F$ are guaranteed to vanish. This implies that we can readily obtain a (larger) lower bound of $2q - (2m_* + 3)$ zero modes. Physically, these zero modes are just linear combinations of different $f$s which decouple from $c$s. Clearly they do not account for the total number of zero modes in the spectrum, i.e., $2q$ at each $k$ or two per moiré unit cell per valley for each spin projection. As we go forth to show, the remaining $2m_* + 3$ zero modes are contributed by the coupled modes, which at $M \neq 0$ get split into a group of states with a width of $2|M|$ accounting for the bandwidth of magnetic subbands within the narrow bands. Below we build a framework for analyzing them.

To that end, we define new fermion fields $\bar{f}$ by the canonical transformation

$$\bar{f}_{b\tau k\bar{r}s} = \sum_{r'=0}^{q-1} V_{\bar{r}r'}f_{b\tau kr's}, \quad (15)$$

where the unitary matrix $V$ is defined via the SVD of matrix $\Upsilon_{mr'} = \sum_{m'=0}^{m_{a\tau}}\sum_{\bar{r}=0}^{q-1}U_{mm'}\Sigma_{m'\bar{r}}V_{\bar{r}r'}$. Here $U$ is a unitary matrix of dimensions $(m_{a,\tau} + 1) \times (m_{a,\tau} + 1)$ and $\Sigma$ is a rectangular matrix of dimensions $(m_{a,\tau} + 1) \times q$ containing the singular values of the matrix $\Upsilon$ along the main diagonal and zeros elsewhere, i.e., $\Sigma_{mr} = \Sigma_m\delta_{mr}$. Moreover, using the closed form expression for $\Upsilon$ stated in the previous section, we find that the matrix $U$ above is extremely close to an identity matrix at low $\mathbf{B}$ (see Supplementary Note 5 for details). Substituting the SVD in Eq. (11) and using $U = \mathbb{I}$, we find that $2q - (2m_* + 3)$ of the $\bar{f}$ modes decouple from the $c$s for each valley, spin and $k$. For example at $\tau = +1$, the modes in Eq. (15) that decouple from the $c$s are the ones with $\bar{r} > m_* + 1$ and

$\bar{r} > m_\star$ for $\bar{f}_{11k\bar{r}s}$ and $\bar{f}_{21k\bar{r}s}$, respectively (see Supplementary Notes 6 and 7 for details at $\tau = +1$ and $\tau = -1$, respectively). We thus recover the $2q - (2m_\star + 3)$ zero modes contributed by the decoupled $f$s as discussed earlier.

For the remaining coupled modes, we note that the sum of the $c$–$c$ and $c$–$f$ couplings in Eqs. (7) and (8) at $\tau = +1$ can be rewritten in the $\bar{f}$ basis as

$$H_{cc}^{+1,s} + \left(H_{cf}^{+1,s} + \text{h.c.}\right) =$$
$$\sum_k \sum_{\alpha,\alpha'=1}^{6} \sum_{m=0}^{m_\alpha} \sum_{m'=0}^{m_{\alpha'}} \langle m|\hat{h}_{\alpha,\alpha'}^{+1,s}|m'\rangle d_{ms\alpha}^\dagger(k) d_{m's\alpha'}(k), \quad (16)$$

where $m_{\alpha=1,\dots,4} = m_{\alpha,+1}$, $m_5 = m_\star + 1$ and $m_6 = m_\star$,

$$d_{ms\alpha}^\dagger(k) = \left(c_{11k0ms}^\dagger, c_{21k0ms}^\dagger, c_{31k0ms}^\dagger, c_{41k0ms}^\dagger, \bar{f}_{11kms}^\dagger, \bar{f}_{21kms}^\dagger\right)_\alpha. \quad (17)$$

The operator $\hat{h}_{\alpha,\alpha'}^{+1,s}$ is defined as

$$\hat{h}_{\alpha,\alpha'}^{+1,s} = \begin{pmatrix} 0 & 0 & -i\sqrt{2}\frac{v_\star}{\ell}\hat{a} & 0 & \gamma\Sigma(\hat{a}^\dagger\hat{a}) & i\sqrt{2}\frac{v_\star'}{\ell}\hat{a}^\dagger\Sigma(\hat{a}^\dagger\hat{a}) \\ 0 & 0 & 0 & i\sqrt{2}\frac{v_\star}{\ell}\hat{a}^\dagger & -i\sqrt{2}\frac{v_\star'}{\ell}\hat{a}\Sigma(\hat{a}^\dagger\hat{a}) & \gamma\Sigma(\hat{a}^\dagger\hat{a}) \\ i\sqrt{2}\frac{v_\star}{\ell}\hat{a}^\dagger & 0 & 0 & M & 0 & 0 \\ 0 & -i\sqrt{2}\frac{v_\star}{\ell}\hat{a} & M & 0 & 0 & 0 \\ \gamma\Sigma(\hat{a}^\dagger\hat{a}) & i\sqrt{2}\frac{v_\star'}{\ell}\Sigma(\hat{a}^\dagger\hat{a})\hat{a}^\dagger & 0 & 0 & 0 & 0 \\ -i\sqrt{2}\frac{v_\star'}{\ell}\Sigma(\hat{a}^\dagger\hat{a})\hat{a} & \gamma\Sigma(\hat{a}^\dagger\hat{a}) & 0 & 0 & 0 & 0 \end{pmatrix}_{\alpha,\alpha'}. \quad (18)$$

Here $\hat{a}$ is a simple h.o. lowering operator with eigenstate $|m\rangle$ and $\Sigma(m) = \Sigma_m$. In the $\mathbf{B} \to 0$ limit, up to the first order in flux, the singular values of $\Upsilon$ can be approximated as $\Sigma(m) \approx 1 - \left(m + \frac{1}{2}\right)\frac{\lambda^2}{\ell^2}$. We note in passing that we do not have to rely on this approximation and can find the full closed form expression for $\Sigma(m)$ as shown in the Eq. (28).

A naive minimal substitution into Eq. (1) with $\lambda = 0$ would reproduce Eq. (18) with unit singular values. However, the decoupling of $2 - (2m_\star + 3)/q$ modes per moiré unit cell per spin in each valley is completely overlooked by the naive minimal substitution. This is the reason why it fails to recover the correct count of subbands within the narrow bands as noted earlier.

While the decoupled $\bar{f}$ modes account for $2q - (2m_\star + 3)$ zero modes, the remaining $2m_\star + 3$ zero modes of the flat band limit (i.e., $M = 0$) originate from the zero modes of the operator in Eq. (18). This can be readily verified via exact solutions, the first of which is a pure $c$-mode

$$\theta_1 = [0, 0, |0\rangle, 0, 0, 0]^T. \quad (19)$$

The above can be interpreted as the anomalous zero-LL of a massless Dirac particle coming from the $c$–$c$ coupling. The remaining spectrum can be solved using the ansätze:

$$\theta_3 = \left[c_1^{(3)}|0\rangle, 0, c_3^{(3)}|1\rangle, 0, c_5^{(3)}|0\rangle, 0\right]^T, \quad (20)$$

$$\theta_5 = \left[c_1^{(5)}|1\rangle, c_2^{(5)}|0\rangle, c_3^{(5)}|2\rangle, 0, c_5^{(5)}|1\rangle, c_6^{(5)}|0\rangle\right]^T, \quad (21)$$

$$\begin{aligned} \theta_{6_m} = & \left[c_1^{(6_m)}|m\rangle, c_2^{(6_m)}|m-1\rangle, c_3^{(6_m)}|m+1\rangle,\right. \\ & \left. c_4^{(6_m)}|m-2\rangle, c_5^{(6_m)}|m\rangle, c_6^{(6_m)}|m-1\rangle\right]^T, \end{aligned} \quad (22)$$

where $m \in \{2, \dots, m_\star + 1\}$. $c_\alpha^{(\mu)}$ denotes the complex coefficient of the corresponding h.o state at index $\alpha$, and $\mu$ labels the ansatz index $\theta_\mu$ (with a slight abuse of notation we have $\mu = 6$ for ansätze in Eq. (22) $\forall m$). Using the above, we can set up the eigen-equation for each $\theta_\mu$, which yields a corresponding decoupled $\mu \times \mu$ Hermitian matrix with eigenvectors $c_\alpha^{(\mu)}$ (see Supplementary Note 6 for details).

The anomalous $c$-mode $\theta_1$ offers one zero mode. The hermitian matrices obtained using ansätze $\theta_3$ and $\theta_5$ offer one zero mode each. The non-zero coefficients of these modes are $c_3^{(3)} = -i\Sigma_0\gamma\ell$, $c_5^{(3)} = \sqrt{2}v_\star$ and $c_3^{(5)} = \Sigma_0\Sigma_1(2v_\star'^2 - \gamma^2\ell^2)$, $c_5^{(5)} = -2i\Sigma_0\gamma\ell v_\star$, $c_6^{(5)} = 2\sqrt{2}v_\star \cdot v_\star'\Sigma_1$, respectively. The hermitian matrix obtained using the ansatz $\theta_{6_m}$ is bipartite and offers two zero modes $\forall m$. The non-zero coefficients for these zero modes are $c_3^{(6_m)} = \sqrt{2}\sqrt{m(m-1)}v_\star'\Sigma_{m-1}$, $c_4^{(6_m)} = i\gamma\ell\sqrt{m+1}\Sigma_{m-1}$, $c_6^{(6_m)} = \sqrt{2}\sqrt{m^2-1}v_\star$ and $c_3^{(6_m)} = -i\gamma\ell\sqrt{m-1}\Sigma_m$, $c_4^{(6_m)} = \sqrt{2}\sqrt{m(m+1)}v_\star'\Sigma_m$, $c_5^{(6_m)} = \sqrt{2}\sqrt{m^2-1}v_\star$, respectively. The coupled modes thus offer a total of $2m_\star + 3$ zero modes. Including the $2q - (2m_\star + 3)$ of the decoupled $f$ modes, at each $k$ we recover sum total of $2q$ zero modes in the non-interacting case for each valley, independent of $m_\star$. This gives the total of 2 states per moiré unit cell per spin independent of $\mathbf{B}$, i.e., the total Chern number 0. Note that the magnetic subbands within the narrow band window remain separated by a gap from the remote subbands for $\mathbf{B} \neq 0$ because the remote bands emanate out of $\mathbf{B} \to 0$ energy eigenvalues $\pm\gamma$ obtained using above ansätze, as shown in Fig. 1.

The analysis can straightforwardly be extended to include interactions using appropriate mean field parameters. In the next sections we illustrate it by discussing the strong coupling Hofstadter spectra at three integer fillings of the narrow bands.

## Electron-electron interactions at integer fillings of narrow bands at $\mathbf{B} \neq 0$

To understand the effect of interactions on single-particle Hofstadter spectra discussed above, we extend our formalism to illustrate the Landau quantization for "one-shot" Hatree-Fock (HF) bands obtained using the mean-field (MF) Hamiltonian for a parent valley polarized (VP) state. The VP state at $\mathbf{B} = 0$ is given by a product of valley polarized $\tilde{f}$-multiplets and the Fermi sea ($|\text{FS}\rangle$) of half-filled $\tilde{c}$ electron bands[36]. The narrow band filling factor $\nu$ then determines the number of $\tilde{f}$ electrons to be filled per moiré unit cell above $|\text{FS}\rangle$. The $U(4)$-flavor of the $\tilde{f}$ electrons to be filled is further dictated by the $U(4)$ Hund's rule discussed in ref. 36. Below, we start with the $\mathbf{B} \neq 0$ solution for single particle charge $\pm 1$ excitation at the charge neutrality point (CNP).

$\boldsymbol{\nu = 0}$: At CNP, the MF interactions for the parent VP state with $\tau = +1$ valley occupied by the $f$ electrons with respect to the spinor in Eq. (17) are taken to be[36]

$$V_{\alpha,\alpha'}^{+1,s,\nu=0} = \begin{pmatrix} 0 & 0 & 0 \\ 0 & -\frac{J}{2}\sigma_0 & 0 \\ 0 & 0 & -\frac{U_1}{2}\sigma_0 \end{pmatrix}_{\alpha,\alpha'}. \quad (23)$$

Within this approximation we continue using the $\mathbf{B} = 0$ MF parameters $J$ and $U_1$ obtained for the parent VP state in ref. 36. The MF parameter $U_1$ is the largest energy scale of the THFM as it corresponds to the strong onsite Coulomb repulsion amidst the localized Wannier states of the $f$ fermions. The MF parameter $J$ corresponds to the energy associated with the ferromagnetic exchange interaction between the $U(4)$ moments of $f$ and $c$ fermions with $a = \{3, 4\}$.

The decoupled $f$ modes in the valley $\tau = +1$ now move to energy $-U_1/2$, while the spectrum of the coupled modes in the same valley can be obtained by solving the eigenvalues of the operator $\hat{h}^{+1,s} + V^{+1,s,\nu=0}$, where $\hat{h}^{+1,s}$ is defined in Eq. (18). The spectrum for sector $\tau = -1$ of the MF Hamitonian is the particle-hole symmetric partner of the spectrum for $\tau = +1$[36]. Thus for a given valley quantum number $\tau$, the $2q - (2m_\star + 3)$ decoupled $f$-modes now move to the energy $-\tau U_1/2$ forming the lower and upper bounds on the strong coupling energy window for narrow bands as shown in the Fig. 2b. In order to understand the mode counting within the narrow band strong coupling energy window, we first discuss the solutions in the flat band limit $M = 0$ which can be obtained using ansätze presented in Eqs. (19)–(22). The anomalous $c$-mode $\theta_1$ in Eq. (19) forms the $\mathbf{B}$

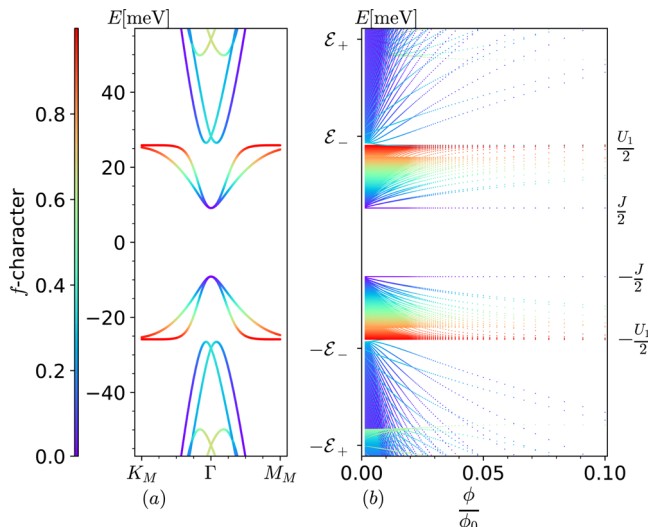

**Fig. 2 | Interacting flat band Hofstadter spectrum at CNP.** The spin degenerate interacting heavy fermion Hofstadter spectrum (**b**) contrasted with zero field strong coupling spectrum (**a**) in flat band ($M = 0$) limit at $w_0/w_1 = 0.7$ for both valleys at CNP. We fix $m_\star = \lceil \frac{q-3}{2} \rceil$. The **B** = 0 energies at $\Gamma$, labeled using $\mathcal{E}_\pm = \pm \frac{U_1}{4} + \sqrt{\frac{U_1^2}{16} + \gamma^2}$, are recovered in **B** → 0 limit of our theory. The value of parameters used are $J = 18.27$ meV, $U_1 = 51.72$ meV, $\gamma = -39.11$ meV, $v'_\star = 1.624$ eV . Å, $v_\star = -4.483$ eV.Å and $\lambda = 0.3792 L_m$. Following ref. 36, twist angle $\theta = 1.05°$ in this work and $L_m = 134.218$ Å. We set $k_{1,2} = 0$, although inconsequential as magnetic subbands are Landau levels in this regime and thus do not disperse. The color labeling represents $f$-character of each energy eigenmode (see caption Fig. 1). The $y$-label $E$[meV] represents energy of subbands in meV.

independent level a $-J/2$ as shown in the Fig. 2b. Using the remaining ansätze $\theta_\mu$, we can set up corresponding $\mu \times \mu$ hermitian matrices with eigenvectors $c^\mu_\alpha$. The three hermitian matrices in total offer $2m_\star + 2$ modes within the strong coupling narrow band energy window. This can be understood by noting that these modes emanate out of the $2m_\star + 2$ fold degenerate **B** → 0 energy eigenvalue $-J/2$ of the above hermitian matrices. Using the fact that $\Sigma_m \to 1$ and $\ell^{-1} \to 0$ as **B** → 0, it can be readily verified that the non-zero coefficients for these **B** → 0 $2m_\star + 2$ degenerate modes at $\tau = +1$ are $c_3^{(3)} = 1$ for $\theta_3$, $c_3^{(5)} = 1$ for $\theta_5$, $c_3^{(6_m)} = 1$ for $\theta_{6_m}$ and $c_4^{(6_m)} = 1$ for $\theta_{6_m}$. Note that in this limit, we have three extra modes of $a = 3$ than the $a = 4$ $c$ fermion. Similarly at $\tau = -1$, we will have three extra modes of $a = 4$ than the $a = 3$ $c$ fermion in the **B** → 0 limit. This can be understood as a direct consequence of a winding number three at $\Gamma$, reported in ref. 29. We present an effective Hamiltonian of these modes in the next paragraph. By the particle-hole symmetry[36], we similarly have $2m_\star + 3$ modes emanating out of $+J/2$ for $\tau = -1$ (see Supplementary Note 7A for details). Including the $2q - (2m_\star + 3)$ decoupled $f$s at energy $-\tau U_1/2$, we have a total of $2q$ magnetic subbands within the narrow band strong coupling window $\pm (J/2$ to $U_1/2)$, i.e., 2 states per moiré unit cell per valley per spin. The remote magnetic subbands on the other hand emanate out of the **B** → 0 energies $-\tau \frac{U_1}{4} \pm \sqrt{\frac{U_1^2}{16} + \gamma^2}$, marked by $\pm \mathcal{E}_{\mp \tau}$ in Fig. 2b. Note that all the **B** → 0 energies mentioned above correspond to the zero field THFM energies at $\Gamma \in$ mBZ at CNP, illustrated in Fig. 2a.

The lowest energy single particle excitations at the CNP at **B** = 0 reside at the $\Gamma$ point, as can be seen in Fig. 2a. The Landau quantization of these bands can be better understood through a simple effective Hamiltonian obtained by systematically projecting onto the subspace spanned by $a = \{3, 4\}c$ fermions. It qualitatively describes the modes emanating from energy eigenvalues $-\tau J/2$. Including both valleys, the effective Hamiltonian for each spin projection is

$$H_{eff} = \begin{pmatrix} H^{\tau=1}_{eff} & 0 \\ 0 & H^{\tau=-1}_{eff} \end{pmatrix}, \text{ where}$$

$$H^{\tau=1}_{eff} = \begin{pmatrix} -\frac{J}{2} - \hbar\omega_c \hat{a}^\dagger \hat{a} & i\frac{A}{\ell^3}\hat{a}^{\dagger 3} \\ h.c. & -\frac{J}{2} - \hbar\omega_c \hat{a}\hat{a}^\dagger \end{pmatrix}, \tag{24}$$

and $H^{\tau=-1}_{eff}$ can be obtained by replacing $\hat{a} \leftrightarrow \hat{a}^\dagger$ and changing the overall sign of $\omega_c$, $A$ and $J$ in $H^{\tau=1}_{eff}$. Also h.c. in Eq. (24) represents hermitian conjugate. The values of coefficient $A$ and effective cyclotron frequency $\omega_c \sim \ell^{-2}$ are provided in the "Effective Hamiltonian coefficients" section. In the **B** → 0 limit we can drop the off-diagonal terms in $H^\tau_{eff}$ because they are $\mathcal{O}\left(\ell^{-3}\right)$. For each spin, the anomalous modes $(|0\rangle, 0)^T$ and $(0, |0\rangle)^T$ at $\tau = +1$ and $-1$, respectively, are singly degenerate at energy $-\tau J/2$. All other modes are doubly degenerate (for each spin). The energies of these pairs are $-\tau(J/2 + n\hbar\omega_c)$ where $n = 1, 2, 3, ...$. Including the spin degeneracy, this would result in a LL filling sequence $0, \pm 2, \pm 6, \pm 10, ...$ in the asymptotic **B** → 0 limit. As **B** increases, however, the off-diagonal terms grow and cause the splitting of these pairs. For example, the splitting of the first pair, i.e., with $n = 1$, is visible at $\phi/\phi_0 \sim 0.025$ (-0.63Tesla) in the Fig. 2b. Moreover, the **B** field required for the splitting of a given pair with an index $n$ decreases with increasing $n$ because each action of the $\hat{a}$ is accompanied by a square root of the LL index making the off-diagonal terms comparable with the diagonal terms at a lower **B**. If we compare the Fig. 2b with the Hofstadter spectrum of the BM model in the strong coupling limit at CNP presented in Fig. 3b, we see a qualitative agreement in the nature of the LL spectra for low **B**. Note that the latter is computed by neglecting the band kinetic energy and using the gauge invariant formalism introduced in the ref. 57 without any recourse to the heavy fermion model. For example, in the vicinity of $\phi/\phi_0 = 0.025$ we can see that the anomalous mode is followed by a nearly degenerate pair of LLs, an isolated LL, and another two nearly degenerate LLs in both Figs. 2b and 3b. Through the effective model analysis presented above we understand that these features appear due to the splitting of asymptotic **B** → 0 degeneracy of non-anomalous modes by the $\mathcal{O}\left(\ell^{-3}\right)$ terms as **B** increases. The splitting amidst the first pair of LLs (after the anomalous mode) appears to grow faster with increasing **B** in Fig. 3b compared to that in Fig. 2b. Thus although the LL sequence at CNP from both approaches is $0, \pm 2, \pm 4$, the LL gap at $\pm 4$ is significantly smaller in the $M = 0$ THFM compared to that in the strong coupling Hofstadter spectrum of BM model when $\phi/\phi_0$ reaches 0.1 (i.e., 2.5 Tesla). Interestingly, the LL filling sequence $0, \pm 2, \pm 4$ at CNP was also reported in the experiment of ref. 42, on an MATBG device with a non-vanishing gap at the CNP at **B** = 0. We come back to the experimental comparison at $\nu = \pm 2$ in the later section.

For $M \neq 0$, the numerically determined strong coupling Hofstadter spectrum for $\tau = -1$, is shown in Fig. 3a (see Supplementary Note 7A for details). As we can see, the lowest mode stays decoupled from the rest of the spectrum. The effect of finite $M$ can be included by adding $M\left(1 - \frac{M_c}{\ell^2}(2\hat{a}^\dagger\hat{a} + 1)\right)\sigma_x \zeta_0$ to $H_{eff}$. The Pauli matrices $\sigma_x$ and $\zeta_0$ act in the $a = \{3, 4\}c$-orbital and valley space, respectively. The value of the coefficient $M_c$ is provided in "Effective Hamiltonian coefficient" section. For non-zero $M$ the double degeneracy of LLs which we saw at $M = 0$ is lifted even in the **B** → 0 limit. This results in a LL sequence $0, \pm 2, \pm 4, ...$ for the parent VP state at CNP for a general **B**.

In the case of parent Kramers intervalley coherent state (KIVC), the effect of finite $M$ is included by adding $-M\left(1 - \frac{M'_c}{\ell^2}(2\hat{a}^\dagger\hat{a} + 1)\right)\sigma_z \zeta_x$ to $H_{eff}$. The coefficient $M'_c$ is presented in the "Effective Hamiltonian coefficients" section. The LLs emanate out of the energy $\pm\sqrt{J^2/4 + M^2}$ and are singly degenerate for each spin projection for a general **B**. Similar to the flat band limit ($M = 0$), the non-anomalous LLs occur in nearly degenerate pairs in the asymptotic **B** → 0 limit (see

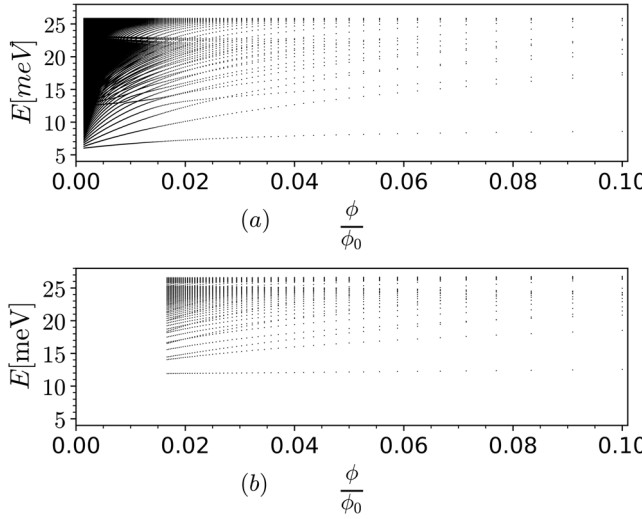

**Fig. 3 | Interacting Hofstadter spectra at CNP. a** THFM Hofstadter spectrum at valley $\mathbf{K}'$ with $M = 3.248$ meV (i.e., including dispersion of the flat bands) for $m_* = \lceil \frac{q-3}{2} \rceil$ and $w_0/w_1 = 0.7$. **b** Strong coupling projected BM Hofstadter spectrum (i.e., in the flat band limit) at $w_0/w_1 = 0.7$ computed using the gauge-invariant basis of magnetic translation group irreps (see Supplementary Note 12 for details). The spectra above are spin degenerate. The $y$-label $E[\text{meV}]$ represents energy of subbands in meV.

Supplementary Fig. 14b). These pairs of LLs split in energy with increasing $\mathbf{B}$. The splitting amidst the first pair of LLs (after the anomalous mode) is much weaker compared to other pairs (see supplementary Fig. 11b). Thus, although the resulting LL sequence is $0, \pm 2, \pm 4$, the LL filling gap $\pm 4$ is much smaller compared to that for $0, \pm 2$ similar to the $M = 0$ case discussed earlier. More details for KIVC can be found in Supplementary Notes 9A and 10A2.

**$\nu = \pm 1$**

In this section, we discuss the Landau quantization of the single particle excitation spectra at the narrow band filling factor of $\nu = -1$ ($\nu = +1$ is related by particle-hole symmetry). Figure 4 shows the $\mathbf{B} = 0$ spectrum and the Hofstadter spectra. We will show that all features of the spectrum can be analytically understood within the formalism, as well as through simple effective models. As in the case of CNP, we continue to use the $\mathbf{B} = 0$ MF interactions for our analysis. The considered MF interactions are computed with respect to a partially spin- and completely valley-polarized parent state. For this state, the valley-spin flavor $\mathbf{K}\uparrow$ for both $b = 1, 2$ ($p_x \pm i p_y$) $f$ fermions and $\mathbf{K}\downarrow$ for $b = 1$ $f$ fermion is occupied (at each unit cell) above the Fermi sea $|\text{FS}\rangle$ of half-filled $c$ fermion bands (see also Supplementary Eq. (S320) in ref. 36).

For the sector valley $\mathbf{K}$ spin $\downarrow$, the charge $\pm 1$ excitations occupy Chern $\mp 1$ bands, which are separated from each other by a sizable gap: the Chern $-1$ and $+1$ bands, marked in red, can be seen in the energy windows $-30$ meV to $-50$ meV and $-55$ meV to $-100$ meV of Fig. 4a, respectively. Below, we elucidate how our formalism captures the fact that the Chern $+ (-)1$ bands gain(lose) states in presence of magnetic field $\mathbf{B}$, as they must to follow the Streda formula[59]. The MF interactions at sector $\mathbf{K}\downarrow$ for the coupled modes with respect to the spinor in Eq. (17) read[36] $V^{+1,\downarrow,\nu=-1}_{\alpha,\alpha'} =$

$$-\begin{pmatrix} W_1 \sigma_0 & 0 & 0 \\ 0 & W_3 \sigma_0 + \frac{J}{2}\sigma_z & 0 \\ 0 & 0 & (U_1 + 6U_2)\sigma_0 + \frac{U_1}{2}\sigma_z \end{pmatrix}_{\alpha,\alpha'}. \quad (25)$$

The MF parameter $W_{a \in \{1,3\}}$ corresponds to the energy associated with the Coulomb repulsion between the $c$ and $f$ fermions, while $U_2$ corresponds to the energy associated with the next nearest neighbor

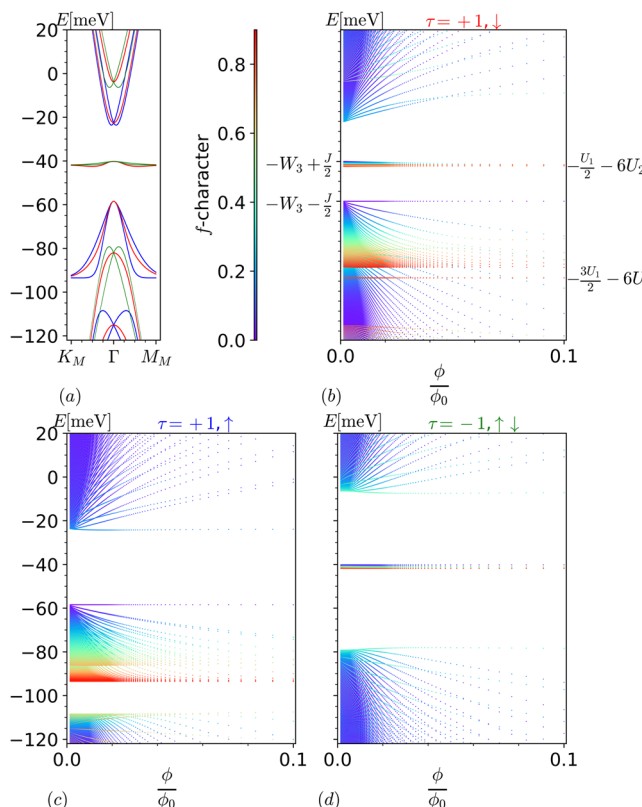

**Fig. 4 | Interacting flat band Hofstadter spectra at $\nu = -1$.** Interacting heavy fermion Hofstadter spectra for sector (**b**) Valley $\mathbf{K}$ spin $\downarrow$, (**c**) Valley $\mathbf{K}$ spin $\uparrow$, and (**d**) Valley $\mathbf{K}'$ spin $\uparrow\downarrow$ (degenerate) contrasted with (**a**) zero field spectrum at filling $\nu = -1$ at $w_0/w_1 = 0.7$, with parameters $W_1 = 44.05$ meV, $W_3 = 49.33$ meV, $U_2 = 2.656$ meV in the flat band limit $M = 0$ with $m_* = \lceil \frac{q-3}{2} \rceil$. The color on panel (**a**) labels the spin and valley sector, red for $\mathbf{K}\downarrow$, blue for $\mathbf{K}\uparrow$, and green for $\mathbf{K}'\uparrow\downarrow$. The color on panels (**b**)–(**d**) labels the $f$-character of each energy eigenmode (see caption Fig. 1). The y-axis in panels **a** and **b**, **c** and **d** are aligned. The $y$-label $E[\text{meV}]$ represents energy of subbands in meV.

Coulomb interaction of the $f$ fermions. The decoupled $f$-modes, i.e., the $\bar{f}_{11kr\downarrow}$ modes with $r \in \{m_* + 2, ..., q-1\}$ and $\bar{f}_{21k\bar{r}\downarrow}$ modes with $\bar{r} \in \{m_* + 1, ..., q-1\}$, are now at energies $-(\frac{3U_1}{2} + 6U_2) = -93.5$ meV and $-(\frac{U_1}{2} + 6U_2) = -41.8$ meV, respectively. The spectrum for the coupled modes can be obtained by solving the eigenvalues of the operator $\hat{h}^{+1,\downarrow} + V^{+1,\downarrow,\nu=-1}$, where $\hat{h}^{+1,s}$ and $V^{+1,\downarrow,\nu=-1}$ are defined in Eqs. (18) and (25), respectively. For $M = 0$, the spectrum is exactly solvable. The anomalous $c$-mode in Eq. (19) is an exact eigenstate which forms the $\mathbf{B}$ independent level at $-(W_3 + \frac{J}{2}) = -58.46$ meV. The remaining spectrum can be solved using the ansätze $\theta_3$, $\theta_5$ and $\theta_{6_m}$, presented in Eqs. (20)–(22). The mode count can be understood as follows:

1. The spectrum for coupled modes includes $m_* + 2$ magnetic subbands emanating out of $\mathbf{B} \to 0$ energy eigenvalue $-(W_3 + \frac{J}{2})$. The non-zero coefficients for these $\mathbf{B} \to 0$ eigenvectors are $c_3^{(3)} = 1$ for $\theta_3$, $c_3^{(5)} = 1$ for $\theta_5$ and $c_3^{(6_m)} = 1$ for $\theta_{6_m}$. Including the anomalous $c$-mode in Eq. (19), we have $m_* + 3$ modes emanating out of $-(W_3 + \frac{J}{2})$. Moreover, accounting the $q - (m_* + 2)$ decoupled $f$ modes at energy $-(\frac{3U_1}{2} + 6U_2)$, we have a total of $q + 1$ magnetic subbands within the energy window of $-55$ meV to $-100$ meV. Recall that the isolated Chern $+1$ band resides in this same energy window at $\mathbf{B} = 0$. We thus see that $q + 1$ magnetic subbands emerge from the Landau quantization of the Chern $+1$ band.

2. The spectrum for coupled modes includes $m_*$ magnetic subbands emanating out of $\mathbf{B} \to 0$ energy eigenvalue $-(W_3 - \frac{J}{2}) = -40.19$ meV. The non-zero coefficients for these $\mathbf{B} \to 0$ eigenvectors are

$c_3^{(6_m)} = 1$ for $\theta_{6_m}$. Including the $q - (m_* + 1)$ decoupled $f$ modes at energy $-(\frac{U_1}{2} + 6U_2)$, we have in total $q - 1$ magnetic subbands in the energy window of $-30$ meV to $-50$ meV. Recall that the isolated Chern $-1$ band resides in this same energy window at $\mathbf{B} = 0$. We thus see that $q - 1$ magnetic subbands emerge from the Landau quantization of the Chern $-1$ band. Through the above mode count analysis, we see that the Chern $\pm 1$ bands Landau quantize into $q \pm 1$ magnetic subbands. Our formalism thus clearly shows that the total number of states per moiré unit cell for the Chern $\pm 1$ bands changes with magnetic field as $1 \pm \frac{1}{q} = 1 \pm \frac{\phi}{\phi_0}$, as expected[59]. The mode count analysis for remaining MF valley-spin sectors, namely valley $\mathbf{K}$ spin $\uparrow$ and $\mathbf{K}'$ spin $\uparrow\downarrow$ (degenerate) can be found in Supplementary Notes 6B, 7B. The Hofstadter spectrum for each MF sector at $\nu = -1$ is shown in Fig. 4b–d for $M = 0$.

To better understand the Landau quantization of the dispersive (light mass) single particle excitations at $\nu = -1$, i.e., in vicinity of $\Gamma$, below we present an effective model analysis similar to that at CNP. As can be seen in Fig. 4d the sectors $\mathbf{K}' \uparrow\downarrow$ contribute magnetic subbands in the energy window $-30$ meV to $-50$ meV; adding a particle into any one of these subbands would move the filling towards CNP. Because we wish to focus on light mass excitations which move the filling away from CNP, we focus on the sectors $\mathbf{K}\uparrow$ and $\mathbf{K}\downarrow$. The effective Hamiltonian at the sector $\mathbf{K}\downarrow$ takes the form

$$H_{eff}^{\nu=-1,\mathbf{K}\downarrow} = \begin{pmatrix} -W_3 - \frac{J}{2} - \hbar\bar{\omega}_c \hat{a}^\dagger \hat{a} & i\frac{\bar{A}}{\ell^3}\hat{a}^{\dagger 3} \\ h.c. & -W_3 + \frac{J}{2} - \hbar\bar{\omega}_c \hat{a}\hat{a}^\dagger \end{pmatrix} + M\left(1 - \frac{\bar{M}_c}{\ell^2}\hat{a}\hat{a}^\dagger - \frac{\bar{M}_c}{\ell^2}\hat{a}^\dagger\hat{a}\right)\sigma_x \quad (26)$$

where Pauli matrix acts in the orbital space of the $a = \{3, 4\}c$ fermions. The coefficients $\bar{A}$, $\bar{M}_c$, $\tilde{M}_c$ and cyclotron frequencies $\tilde{\omega}_c \sim \ell^{-2}$, $\bar{\omega}_c \sim \ell^{-2}$ are provided in the "Effective Hamiltonian coefficients" section. The magnetic subbands of interest (the ones emerging from the Landau quantization of the light mass excitations) emanate out of the energy $-W_3 - \sqrt{\frac{J^2}{4} + M^2} = -59.03$ meV, and are singly degenerate for a general $\mathbf{B}$.

The effective Hamiltonian at sector $\mathbf{K}\uparrow$ takes the same form as in Eq. (24). It can be obtained by replacing $-\frac{J}{2}$, $\omega_c$ and $A$ by $-(W_3 + \frac{J}{2})$, $\bar{\omega}_c$ and $\bar{A}'$, respectively, in Eq. (24). The effect of $M$ is included by adding $M\left(1 - \frac{\bar{M}_c'}{\ell^2}(2\hat{a}^\dagger\hat{a} + 1)\right)\sigma_x$ to the above obtained effective Hamiltonian. The coefficients $\bar{A}'$ and $\bar{M}_c'$ are provided in the "Effective Hamiltonian coefficients" section. The LLs emerge out of energies $-(W_3 + J/2) + M = -55.22$ meV and $-(W_3 + J/2) - M = -61.71$ meV, and are singly degenerate for a general $\mathbf{B}$.

In the $\mathbf{B} \to 0$ limit, we can drop the off-diagonal $\mathcal{O}\left(\ell^{-3}\right)$ term in both of the above effective Hamiltonians. Further taking the flat band limit $M = 0$, we find that LL energies take the form $-(W_3 + \frac{J}{2} + n\hbar\bar{\omega}_c)$, with $n \in \{0, 1, 2, ...\}$. The $\mathbf{B}$ independent anomalous mode ($n = 0$) is doubly degenerate as it is part of the spectrum at both sectors. The remaining modes ($n > 0$) are triply degenerate each: singly degenerate at sector $\mathbf{K}\downarrow$ and doubly degenerate at sector $\mathbf{K}\uparrow$. This results in the LL sequence of $+1, -1, -4, -7, ...$ for $M = 0$ in asymptotic $\mathbf{B} \to 0$ limit. The $+1$ gap in the sequence appears due to the Chern number $+1$ of the occupied band at sector $\mathbf{K}\downarrow$ at $\mathbf{B} = 0$.

Relaxing the $\mathbf{B} \to 0$ limit above, i.e., including the $\mathcal{O}\left(\ell^{-3}\right)$ terms in the above effective Hamiltonians (still $M = 0$), we see that the LL energies change to: $E_m$, $E_n^\downarrow$ and $E_m$, $E_n^{\uparrow \pm}$ at sectors $\mathbf{K}\downarrow$ and $\mathbf{K}\uparrow$, respectively. Here $m \in \{0, 1, 2\}$, $n \in \{0, 1, 2, ...\}$, $E_m = -W_3 - J/2 - m\hbar\bar{\omega}_c$, $E_n^\downarrow = -W_3 - 3\hbar\bar{\omega}_c/2 - \hbar\tilde{\omega}_c/2 - f_n - \sqrt{u_n^2 + v_n\bar{A}^2}$ and $E_n^{\uparrow \pm} = -(W_3 + J/2) - \hbar\bar{\omega}_c(n + 2) \pm \sqrt{\hbar^2\bar{\omega}_c^2 + v_n\bar{A}'^2}$. The coefficients $f_n = n\hbar(\bar{\omega}_c + \tilde{\omega}_c)/2$, $u_n = J/2 + 3\hbar\bar{\omega}_c/2 - \hbar\tilde{\omega}_c/2 + n\hbar(\bar{\omega}_c - \tilde{\omega}_c)/2$ and $v_n = (n + 1)(n + 2)(n + 3)/$

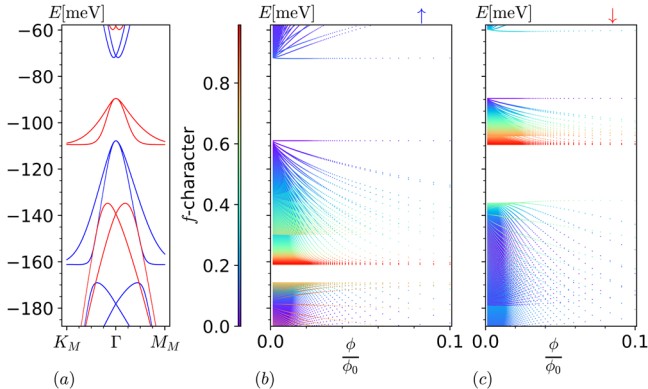

**Fig. 5 | Interacting flat band Hofstadter spectra at $\nu = -2$.** Interacting heavy fermion Hofstadter spectra for sectors (**b**) valley $\mathbf{K}$ spin $\uparrow$ and (**c**) valley $\mathbf{K}$ spin $\downarrow$ contrasted with the (**a**) zero-field spectrum at filling $\nu = -2$ at $w_0/w_1 = 0.7$ in the flat band limit $M = 0$ with $m_* = \lceil\frac{q-3}{2}\rceil$. The $y$-axis in all the above three panels are aligned. The color on panel (**a**) labels the spin sector at valley $\mathbf{K}$, blue and red for spin $\uparrow$ and spin $\downarrow$, respectively. The color on panels (**b**) and (**c**) labels the $f$-character of each energy eigenmode (see caption Fig. 1). The $y$-label $E$[meV] represents energy of subbands in meV.

$\ell^6$. Thus apart from the doubly degenerate levels at $E_m$, all others levels are singly degenerate. Since the LL energies are in the order: $E_0 > E_0^{\uparrow,+} > E_1 > E_1^{\uparrow,+} > ...$, the resulting LL sequence is $+1, -1, -2, -4, -5, ...$. Upon relaxing the flat band limit, i.e., for $M \neq 0$, all the above LL degeneracies get lifted. This results in LL sequence $+1, 0, -1, ...$ for a general $\mathbf{B}$, with LL gaps at $+1, -1$ being the most dominant.

## $\nu = \pm 2$

In this section, we discuss the Landau quantization of the single particle excitation spectra at the narrow band filling factor of $\nu = -2$ ($\nu = +2$ is related by particle-hole symmetry). Figure 5 shows the $\mathbf{B} = 0$ spectrum and its continuation in field. As before, we will also derive the dominant LL sequence using an effective model that is simple enough for analytical solutions while capturing the low-energy features. As in the previous sections, we continue to use the $\mathbf{B} = 0$ MF interactions for our analysis. The considered MF interactions are computed with respect to a spin and valley polarized parent state, for which the valley-spin flavor $\mathbf{K}\uparrow$ for both $b = 1, 2$ ($p_x \pm ip_y$) $f$ fermions are occupied (at each unit cell) above the Fermi sea $|\text{FS}\rangle$ of half-filled $c$ fermion bands (see also Supplementary Eq. (S333) in ref. 36).

Below we discuss the valley sector $\mathbf{K}$ of the resulting MF Hamiltonian, for both spin $s = \uparrow$ and $\downarrow$. The single particle charge $+1$ excitations occupy the red band in energy window $-89.5$ meV to $-109.5$ meV of the Fig. 5a, which is part of the spectrum at sector $\mathbf{K}\downarrow$. On the other hand, the single particle charge $-1$ excitations occupy the dispersive blue band in energy window $-107.8$ meV to $-161.2$ meV of the Fig. 5a, which is part of the spectrum at sector $\mathbf{K}\uparrow$. The MF interactions for spin $s$, with respect to the spinor in Eq. (17) reads[36] $V_{\alpha,\alpha'}^{+1,s,\nu=-2} =$

$$-2\begin{pmatrix} W_1\sigma_0 & 0 & 0 \\ 0 & (W_3 + \zeta_s\frac{J}{4})\sigma_0 & 0 \\ 0 & 0 & \left(\frac{4+\zeta_s}{4}U_1 + 6U_2\right)\sigma_0 \end{pmatrix}_{\alpha,\alpha'}, \quad (27)$$

where $\zeta_s = (+) -1$ for $s = (\uparrow) \downarrow$. For the spin $s$, the $2q - (2m_* + 3)$ decoupled $f$ modes are at energy $-\left(\frac{4+\zeta_s}{2}U_1 + 12U_2\right)$, i.e., $(-161.17$ meV) $-109.45$ meV for $s = (\uparrow) \downarrow$. The spectrum for the coupled modes can be obtained by solving the eigenvalues of the operator $\hat{h}^{+1,s} + V^{+1,s,\nu=-2}$, where $\hat{h}^{+1,s}$ and $V^{+1,s,\nu=-2}$ are defined in Eqs. (18) and (27), respectively. In the flat band limit $M = 0$, the

spectrum for coupled modes is exactly solvable. The anomalous $c$-mode in Eq. (19) is an exact eigenstate which forms the **B** independent level at $-(2W_3 + \zeta_s \frac{J}{2})$, i.e., $(-107.79\,\text{meV}) - 89.52\,\text{meV}$ for $s = (\uparrow) \downarrow$. The remaining spectrum can be solved using the ansätze $\theta_3$, $\theta_5$, and $\theta_{6_m}$, presented in Eqs. (20)–(22). The spectrum of coupled modes include $2m_\star + 2$ magnetic subbands emanating out of the $\mathbf{B} \to 0$ energy eigenvalue $-(2W_3 + \zeta_s \frac{J}{2})$. The non-zero coefficients for the corresponding $\mathbf{B} \to 0$ eigenvectors are $c_3^{(3)} = 1$ for $\theta_3$, $c_3^{(5)} = 1$ for $\theta_5$, $c_3^{(6_m)} = 1$ for $\theta_{6_m}$ and $c_4^{(6_m)} = 1$ for $\theta_{6_m}$. We thus have a total of $2m_\star + 3$ magnetic subbands emanating out of $\mathbf{B} \to 0$ energy $-(2W_3 + \zeta_s \frac{J}{2})$. Including the $2q - (2m_\star + 3)$ decoupled $f$ modes at $-\left(\frac{4+\zeta_s}{2} U_1 + 12U_2\right)$, we have a total of $2q$ magnetic subbands in the energy window $\left(-(2W_3 + \zeta_s \frac{J}{2}) \text{ to } -(\frac{4+\zeta_s}{2} U_1 + 12U_2)\right)$, i.e., $-89.52$ meV to $-109.45$ meV and $-107.79$ meV to $-161.17$ meV for $s = \downarrow$ and $\uparrow$, respectively. Recall that the red ($\downarrow$) and blue ($\uparrow$) bands discussed earlier reside in the same energy window at $\mathbf{B} = 0$. We thus have a total of $2q$ magnetic subbands emerging from the Landau quantization of single particle charge $\pm 1$ excitation bands at valley sector $\mathbf{K}$, i.e., two states per moiré unit cell for each. The discussion for sector $\mathbf{K}' \downarrow\uparrow$ can be found in Supplementary Note 7C.

To better understand the Landau quantization of the dispersive (light mass) single particle excitations at $\nu = -2$, i.e., in vicinity of $\Gamma$, below we present an effective model analysis similar to that in previous sections. As can be seen in the Fig. 5c, the sector $\mathbf{K}\downarrow$ contributes to magnetic subbands in the energy window $-89.52$ meV to $-109.45$ meV; adding a particle into any one of these subbands would move the filling towards CNP. Same is true for the magnetic subbands contributed by sectors $\mathbf{K}' \uparrow\downarrow$, as can be seen in the Supplementary Fig. 9. Because we wish to focus on light mass excitations which move the filling away from CNP, we focus only on the sector $\mathbf{K}\uparrow$.

The effective Hamiltonian at sector $\mathbf{K}\uparrow$ takes the same form as in Eq. (24). It can be obtained by replacing $-\frac{J}{2}$, $\omega_c$ and $A$ by $-(2W_3 + \frac{J}{2})$, $\omega_c^{(\uparrow)}$ and $A^{(\uparrow)}$, respectively, in Eq. (24). The effect of $M$ is included by adding $M\left(1 - \frac{M_c^{(\uparrow)}}{\ell^2}(2\hat{a}^\dagger \hat{a} + 1)\right)\sigma_x$ to the above obtained effective Hamiltonian. The coefficients $\omega_c^{(\uparrow)}$, $A^{(\uparrow)}$ and $M_c^{(\uparrow)}$ are provided in the "Effective Hamiltonian coefficients" section. The LLs emanate out of the energy $-(2W_3 + J/2) \pm M$ and are singly degenerate for a general $\mathbf{B}$. In the $\mathbf{B} \to 0$ limit, we can drop the off-diagonal $\mathcal{O}(\ell^{-3})$ terms in the above effective Hamiltonian. Further setting $M = 0$, we see that the LL energies take the form $-(2W_3 + \frac{J}{2} + n\hbar\omega_c^{(\uparrow)})$, with $n \in \{0, 1, 2...\}$. Similar to CNP, except the anomalous mode ($n = 0$) at energy $-(2W_3 + \frac{J}{2})$, every remaining mode ($n > 0$) is doubly degenerate. This results in the LL sequence $-1, -3, -5, \ldots$ for $M = 0$ in the asymptotic $\mathbf{B} \to 0$ limit. This asymptotic degeneracy of the non-anomalous modes is lifted as $\mathbf{B}$ increases, as is seen when $\mathcal{O}(\ell^{-3})$ terms are included. Relaxing the flat band limit ($M \neq 0$) lifts the double degeneracy of the non-anomalous modes even in the asymptotic $\mathbf{B} \to 0$ limit. Thus for $M \neq 0$, we have the LL sequence $-1, -2, -3, \ldots$ for a general $\mathbf{B}$. Contrary to $\nu = 0$, the LL sequence obtained at $\nu = -2$ differs from the LL filling sequence $-2, -4, -6$ reported in the experiment of ref. 42 at $\nu = -2$ by the appearance of the sizable gap at $-1$. Studying the origin of this difference will be a subject of future work.

## Discussion

We have put forward a generalization of THFM in finite $\mathbf{B}$. Although the formalism applies to any rational value of $\frac{\phi}{\phi_0}$, the physical nature of hybridization amidst the heavy $f$ and topological $c$ fermions is particularly revealing for the $\frac{1}{q}$ sequence. The finite $\mathbf{B}$ analytical solution in the flat band limit provides an intuitive picture of the mechanism for Landau quantization of the strong coupling spectra of MATBG at integer fillings in terms of the decoupled $f$ modes and coupled $c$–$f$ modes, all the way to zero magnetic field. It also provides a deeper understanding of the nature of the $\pm\frac{J}{2}$ level at CNP, observed in numerics before[56], as the anomalous zero-LL of a massless Dirac particle, a key ingredient of the topological heavy fermion

picture of MATBG. Although the number of the decoupled $f$-modes per unit cell per spin at CNP is dependent on the LL index upper cutoff, the total number of states in the narrow band strong coupling window remains pinned to 2 per unit cell per spin, independent of the upper cutoff, as expected for a total Chern number 0. Even though the full $M \neq 0$ problem requires numerical analysis, we are able to probe till fluxes at least as low as 1/700, which was not possible through the the framework of strong coupling expansion. We moreover argue that the overall physical nature of the subbands should stay unchanged, as $M$ anyways is the smallest energy scale in the problem. Although we present the Landau quantization of one-shot HF bands in order to outline the theoretical procedure, in practice one can use the same methodology to Landau quantize the self-consistent HF bands. We argue that it would not drastically alter any of the interacting Hofstadter spectrum features because the one-shot states are adiabatically connected to the self-consistent states, owing to almost identical band structure features as the self-consistent state at every $\nu$ discussed[36]. Throughout the text we neglected the spin Zeeman effect as it leads to a much smaller energy splitting than the orbital effect, the former is only a few Kelvin at the highest fields considered here while the latter is several meV, so at least an order of magnitude larger. The effect of renormalization of mean field parameters in magnetic field and heterostrain is yet to be incorporated in our framework. A full analysis for other integer fillings, translation symmetry broken candidate ground states and Hofstadter-scale fluxes where reentrant many-body and topological effects are at play[52,66–68], is also left for the future work.

## Methods

### Evaluation of the $c$–$f$ matrix elements at $\mathbf{B} \neq 0$

Because $\eta_{b\tau kr}(\mathbf{r})$ is constructed using repeated action of MT operators on the Wannier state $W_{0,b\tau}(\mathbf{r})$ as defined in Eq. (3), and because the MT operators commute with $H_{BM}^\tau(\mathbf{p} - \frac{e}{c}\mathbf{A})$, we can reorder them so that $H_{BM}^\tau(\mathbf{p} - \frac{e}{c}\mathbf{A})$ acts directly on $W_{0,b\tau}(\mathbf{r})$. Since $H_{BM}^\tau(\mathbf{p} - \frac{e}{c}\mathbf{A})$ is linear in $\mathbf{p} - \frac{e}{c}\mathbf{A}$, the vector potential $\mathbf{A}$ now acts on the well localized state $W_{0,b\tau}(\mathbf{r})$ centered at the origin where $\mathbf{A}$ vanishes. Therefore, even though $\mathbf{A}$ is large at large $x$, we can safely neglect its contribution at low $\mathbf{B}$ (confirmed numerically in Supplementary Note 4C). Moreover, since $\Psi_{a\tau}^\dagger(\mathbf{r})$ is a Bloch state at $\Gamma$, it is invariant under the action of the moiré lattice translation operators. The $c$–$f$ coupling $h_{[amr],[br']}^\tau(k)$ at low $\mathbf{B}$ thus reduces to calculating the integral $\int d^2\mathbf{r}\left(\hat{t}_{\mathbf{L}_2}^{-n}\hat{t}_{\mathbf{L}_1}^{-s}\chi_{krm}(\mathbf{r})\right)^\dagger \Psi_{a\tau}^\dagger(\mathbf{r}) H_{BM}^\tau(\mathbf{p}) W_{0,b\tau}(\mathbf{r})$, summed over all integer values of $s$ and $n$ and weighted by the factor of $e^{2\pi i(sk_1 + n(k_2 + r'/q))}/\sqrt{\mathcal{N}}$ as follows from the Eq. (3). The factor $\Psi_{a\tau}^\dagger(\mathbf{r}) H_{BM}^\tau(\mathbf{p}) W_{0,b\tau}(\mathbf{r})$ involves the Hamiltonian as well as the $c$ and $f$ wavefunctions strictly at $\mathbf{B} = 0$. Its Fourier transform was calculated in ref. 36 and sets the $c$–$f$ coupling at $\mathbf{B} = 0$ in momentum space, $e^{-\frac{1}{2}k^2\lambda^2} H^{cf,\tau}(\mathbf{k})$, appearing in the Eq. (1). Therefore, inverse Fourier transforming it gives $\Psi_{a\tau}^\dagger(\mathbf{r}) H_{BM}^\tau(\mathbf{p}) W_{0,b\tau}(\mathbf{r}) = \sqrt{A_{uc}} H^{cf,\tau}(-i\boldsymbol{\nabla}_\mathbf{r}) e^{-\frac{r^2}{2\lambda^2}}/(2\pi\lambda^2)$. The factor $\hat{t}_{\mathbf{L}_2}^{-n}\hat{t}_{\mathbf{L}_1}^{-s}\chi_{krm}(\mathbf{r})$ can be computed by noting that $\chi_{krm}(\mathbf{r})$ is an eigenstate of $\hat{t}_{\mathbf{L}_1}$ and $\hat{t}_{\mathbf{L}_2}^{-n}\chi_{krm}(\mathbf{r}) = e^{2\pi i n(k_2 + r/q)}\chi_{[k_1 + n\frac{p}{q}],k_2 m}(\mathbf{r})$. Finally substituting the explicit expression of $\chi_{[k_1 + n\frac{p}{q}],k_2 m}(\mathbf{r})$ using Eq. (4) reduces $h_{[amr],[br']}^\tau(k)$ to sum over integrals of a 2D gaussian, a plane wave factor along $y$, and shifted 1D h.o. wavefunctions along $x$. We thus have a standard gaussian integral in $y$, while the $x$-integral over the gaussian and and shifted 1D h.o. wavefunctions can be evaluated using results in ref. 64 (see Supplementary Note 4B for details).

It is particularly revealing to analyze the case $p = 1$, i.e., the $\phi/\phi_0 = 1/q$ sequence. Since $r$ ranges from 0 to $p - 1$, the $1/q$ sequence is tantamount to setting $r = 0$ in $h_{[amr],[br']}^\tau$. Based on the results from the above discussion, after performing the summation over $n$, we find that

$h^{\tau}_{[am0],[br']}(k)$ reduces to $\int d^2\mathbf{r}\left(\hat{t}^{r'+jq}_{L_1}\Phi_m(\mathbf{r},k_2\mathbf{g}_2)\right)^* H^{cf,\tau}_{ab}(-i\boldsymbol{\nabla}_\mathbf{r})e^{-\frac{r^2}{2\lambda^2}}$, summed over all integer values of $j$ and weighted by the factor $e^{-2\pi i(r'+jq)k_1}\sqrt{L_{1x}/\ell}/(2\pi\lambda^2)$. For $a=b=1$, this integral can be visualized as an overlap between a 2D localized heavy state with size $\lambda$ sitting at the origin and a 1D h.o. shifted in the $x$-direction with a plane wave phase variation in the $y$-direction that depends on the shift (see Supplementary Fig. 2). To understand for what choice of $m,r',j$ is this integral significant, note that the h.o. wavefunction is localized in the $x$ direction about $(r'+jq+k_2q)L_{1x}$, and its width is $\sim 2\sqrt{2m+1}\sqrt{q}$ unit cells. In addition, the combination $\frac{r'}{q}+j+k_2$ controls the period of oscillation in the $y$-direction set by $1/(\frac{r'}{q}+j+k_2)$ times the unit cell size. The integer $r'+jq$ thus determines the unit cell to which the h.o. is shifted, and, because $k_2 2\pi\ell^2/L_m=k_2qL_{1x}$, the value of $k_2q\in[0,1)$ fine-tunes the shift within the unit cell. The index $j$ then determines $q$-unit-cell periodic revival of the h.o. states, also illustrated in Supplementary Fig. 2. Consider the case $r'=j=0$. The h.o. is centered at the unit cell containing the localized heavy state and the period of oscillations in the $y$-direction is long compared to the unit cell, encompassing at least $q$ unit cells. The hybridization with the localized heavy state proportional to $\gamma$ is thus significant. The spatial extent of the h.o. state in the $x$-direction is $\sim 2\sqrt{2m+1}\sqrt{q}$ unit cells, which at low $\mathbf{B}$ is much longer than the localized heavy state. Thus unless $m$ is close to $m_\star \lesssim q/2$, even though the h.o. state oscillates and has $m$-nodes, the result of the integration will be approximately given by the value of the h.o. wavefunction at $-k_2qL_{1x}$, up to an overall phase. If we keep $j=0$ but increase $r'$ to 1 then the h.o. is centered at the unit cell adjacent to the one containing the localized heavy state and the period of oscillations in the $y$-direction is still long, between $q/2$ and $q$ unit cells. The hybridization with the localized heavy state proportional to $\gamma$ will still be significant and the result of the integration will still be approximately given by the value of the h.o. wavefunction but now at $-(1+k_2q)L_{1x}$, up to an overall phase. However for values of $r'$ past $\sim\sqrt{2m+1}\sqrt{q}$, regardless of the value of $qk_2$, the contribution from the revival copy $j=0$ gets exponentially suppressed, due to the large off-set with the 2D localized state. So for values of $r'>q/2$, it is the $j=-1$ revival copy of the h.o. states which gives the dominant contribution. We thus neglect all other values of $j$ and only consider the contribution from h.o. state centered at the unit cell $r'+jq\to \mathrm{sgn}_+\left(\frac{q}{2}-r'\right)\min[r',q-r']$, where $\mathrm{sgn}_+(x)$ is the usual sign function except at 0 where it evaluates to 1. The $\mathcal{F}_m$ appearing in the compact expression of $c-f$ hybridization in Eq. (13) comes from the $x$-integral, i.e., overlap of the 2D heavy localized state with harmonic oscillator wavefunction $\varphi_m$. In the limit $\lambda\to 0$, the 2D heavy localized state becomes the Dirac $\delta$-function, and we recover $\mathcal{F}_m(\lambda\to 0,x_0)=\varphi_m(-x_0)$ as expected. We found that keeping the full form of $\mathcal{F}_m$ is needed in order to achieve accurate results even for the low $B$ range, therefore we do not take this limit when handling $\mathcal{F}_m$ (see also Supplementary Fig. 4). The exponential suppression factor multiplying $\mathcal{F}_m$ in Eq. (13) comes from the $y$-integral; its dependence on $r'_q$ is weaker than $\mathcal{F}_m$, which comes from the $x$-integral.

The derivatives appearing in case of the matrix elements $h^{\tau}_{[1(2)m0],[2(1)r']}$, act on the localized heavy function to change its spatial symmetry from $s$ to $p_{x,y}$-like. Moreover an integration by parts and expressing the derivatives via h.o. raising and lowering operators allows us to relate these cases to the analysis without derivatives (see Supplementary Notes 4B2, 4B3 and 4C for details).

**Closed form expression for the singular values of $\Upsilon$ appearing in Eq. 18**

The fact that $U$ is very close to an identity matrix allows us to obtain an analytical expression for $\Sigma(m)$, which reads

$$\Sigma(m)=\left(\frac{1}{\sqrt{\xi(\kappa)}}\frac{1}{2^m m!}\mathcal{H}_{mm}\left(0,\frac{\kappa^6}{\xi(\kappa)};0,\frac{\kappa^6}{\xi(\kappa)}\bigg|\frac{2}{\xi(\kappa)}\right)\right)^{\frac{1}{2}},\quad (28)$$

where $\kappa^2=\frac{\lambda^2}{\ell^2}=(\frac{\phi}{\phi_0})2\pi\lambda^2/(L_{1x}L_m)$, $\xi(\kappa)=(1+\kappa^2+\kappa^4)(1+\kappa^2)$ and

$$\mathcal{H}_{mn}(x,y;w,z|\beta)=\sum_{k=0}^{\min(m,n)}\frac{m!n!\beta^k}{(n-k)!(m-k)!k!}\mathcal{H}_{m-k}(x,y)\mathcal{H}_{n-k}(w,z).$$

$$(29)$$

Further details of the derivation can be found in Supplementary Note 5.

**Effective Hamiltonian coefficients**

The coefficients appearing in the effective Hamiltonian at CNP in flat band limit presented in Eq. (24) are $\ell^2\hbar\omega_c=4.00\times 10^5$ meV Å$^2$ and $A=4.27\times 10^7$ meV Å$^3$. The coefficient appearing in the mass term for VP and KIVC states at CNP are $M_c=1.28\times 10^4$ Å$^2$ and $M'_c=2.09\times 10^4$ Å$^2$, respectively. The coefficients appearing in the effective Hamiltonian given in Eq. (26) are $\ell^2\hbar\bar\omega_c=6.92\times 10^5$ meV Å$^2$, $\ell^2\hbar\tilde\omega_c=4.22\times 10^4$ meV Å$^2$, $\bar{A}=6.69\times 10^7$ meV Å$^3$, $\bar{M}_c=1.54\times 10^4$ Å$^2$ and $\tilde{M}_c=2.54\times 10^4$ Å$^2$. The coefficients appearing in the effective Hamiltonian for sector $\mathbf{K}\uparrow$ at $\nu=-1$ are $\bar{A}'=6.11\times 10^7$ meV Å$^3$ and $\bar{M}'_c=1.34\times 10^4$ Å$^2$. The coefficients appearing in the effective Hamiltonian for sector $\mathbf{K}\uparrow$ at $\nu=-2$ are $\ell^2\hbar\omega^{(\uparrow)}_c=8.31\times 10^5$ meV Å$^2$, $A^{(\uparrow)}=5.79\times 10^7$ meV Å$^3$, and $M^{(\uparrow)}_c=1.32\times 10^4$ Å$^2$. The derivation of the effective Hamiltonians and full expressions for the coefficients can be found in Supplementary Notes 10A, 10B and 10C.

**Reporting summary**

Further information on research design is available in the Nature Portfolio Reporting Summary linked to this article.

# Data availability

The data needed to evaluate the conclusions in the paper are present in the paper and the Supplementary Material.

# Code availability

The code used to generate the figures in the paper has publicly been made available by the authors at repository.

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

## Acknowledgements

The authors thank Xiaoyu Wang and Zhi-da Song for valuable conversations, and to Dumitru Călugăru for computational advice. J.H.-A. is supported by a Hertz Fellowship and by ONR Grant No. N00014-20-1-2303. B.A.B. is supported by the DOE Grant No. DE-SC0016239 and by the EPiQS Initiative, Grant GBMF11070. A.C. was supported by Grant No. GBMF8685 towards the Princeton theory program and by the Gordon and Betty Moore Foundation through the EPiQS Initiative, Grant GBMF11070. Further sabbatical support for A.C., J.H.A. and B.A.B. was provided by the European Research Council (ERC) under the European Union's Horizon 2020 research and innovation program (grant agreement No. 101020833), the Schmidt Fund for Innovative Research, Simons Investigator Grant No. 404513. O.V. was supported by NSF Grant No. DMR-1916958 and is partially funded by the Gordon and Betty Moore Foundation's EPiQS Initiative Grant GBMF11070, National High Magnetic Field Laboratory through NSF Grant No. DMR-1157490 and the State of Florida.

## Author contributions

K.S., A.C., J.H.A., B.A.B., and O.V. all contributed to the intellectual content of the work.

## Competing interests

The authors declare no competing interests.
