## [Peer Review File · Nature Communications]

Topological heavy fermions in magnetic fieldREVIEWER COMMENTS

Reviewer #1 (Remarks to the Author):

The manuscript "Topological Heavy Fermions in magnetic Field" examines the problem of how to construct the Landau level (LL) spectrum of eigenstates for the narrow bands of twisted bilayer graphene (TBG) in a magnetic field. It does so within the so-called "topological heavy fermion" (THF) approach pioneered by Song and Bernevig, in which the topological obstruction to building localized states that capture the flat bands is circumvented by viewing the bands as emerging from the hybridization of well-localized f-electrons and delocalized c-electrons. In this picture, the f-electron levels (being well-localized) do not directly see the magnetic field, whereas the c-electrons form Landau levels linked to those of weakly gapped Dirac fermions (or gapless ones in the flat-band limit). The challenge is to understand the LL spectrum of TBG which comes from hybridizing these bands; since the total number of f-states is B-independent whereas the c-electron levels have the usual LL degeneracy there must be some nontrivial coupling to maintain this structure, which the authors argue emerges from an intricate pattern in the c-f coupling (that goes beyond a "minimal coupling" picture of the f-electron levels). They show how to build the appropriate model and apply it to understanding the physics at a variety of fillings, along the way shedding light on the nature of a previously-numerically-observed (almost-)dispersionless (in B) LL as linked to "anomalous" level of the c-electrons viewed as (almost) gapless Dirac electrons. A key innovation is the analytical control afforded by the authors' calculations, especially in the flat band limit, that lends insight into what previously only accessibly numerically.

The work is clearly of significance to anyone working to understand TBG using the lens of the heavy-fermion picture: while originally primarily collaborators of some of the present authors, an increasing number of groups are adopting this approach, particularly to study relatively high-energy/high-temperature processes such as "cascade" physics and photoemission. It is also clearly of relevance to understanding other twisted graphene systems with a similar heavy-fermion description. It is clearly a worthy and valuable addition to the established literature that fully supports its claims and provides exhaustive detail.

In fact, while overall I ultimately lean to recommending publication, my main concern stems from the technical level of the work -- it is fairly heavy going even for experts, and the pedagogy, while refreshingly direct and minutely detailed, occasionally can lead to a loss of the "big picture" due to the weighty formalism that is (rightly) introduced to solve the problem. Therefore, while I think that overall the work is very good, my recommendation would be to improve the presentation to add pedagogical material and some more pointers to the uninitiated. Some of the comments/questions below offer suggestions for such improvements-- while others are more narrow technical inquiries/minor typos that I caught.

(1) I think it would be very helpful for the authors to give an overview of why just minimal coupling the f-electrons won't work, to introduce the reader to the key challenge addressed here (maybe a good place would be the third paragraph of the introduction).

(2) I just want to confirm that the notation in eq(5) is consistent with Ref[39] -- there the J and U_1 are linked to quartic terms, but I took eq (5) (per the statement "mean field interaction") to just come from a replacement of these with the appropriate quadratic mean-field term -- is this correct?

(3) Also in the intro, the authors describe the anomalous almost B-independent mode, in terms of the zero-LL of a massless Dirac particle. My understanding is that this is true only for exactly flat bands ($M=0$), and that there would be a weak dependence for non-zero M . Is this correct? I mention this because otherwise I don't see a zero-energy Dirac fermion in the problem -- I thought that for $M \neq 0$ there was only a quadratic band crossing?

(4) In Fig. 1, 2, I think the spectrum is shown for both valleys, correct? If not, I don't immediately see how one can get a level at both $\pm J/2$ on the basis of Eq. (5). It might be worth flagging in different cases when one is talking about single-valley spectra and where one is discussing the both valleys (and both spins, where relevant)

(5) I think it would be worth crisply defining that "flat band" is equivalent to " $M=0$ " somewhere and that this is the terminology use throughout. Often people colloquially use flat and almost-flat interchangeably as in the "flat bands of moire graphene" even when the flatness is not exact. My understanding is that the authors are being (appropriately) more precise, but it's good to say this.

(6) What is the meaning of p in equations (6) and (7) -- it does not appear on the RHS, so it's a bit confusing. I suspect that perhaps by p here one means the appropriate operator identification $p \rightarrow -i \hbar \nabla$, but I might have misunderstood this completely. It is not central to the discussion but would be good to clarify.

(7) When discussing the degeneracy of each LL (line 199) it might be good to remind the reader that $N\phi$ is the total flux through the system in the authors' convention, so that it's the familiar expression for the degeneracy.

(8) From eq(20), is it correct to conclude that the explicit dependence of A is unimportant to the c - f coupling, but only the implicit structure deduced from the structure of the LL wavefunctions of c - and f - electrons? If so it's worth saying so explicitly.

(9) The SVD approach is very elegant but could perhaps be summarized more crisply.

In summary, I believe this is a very good paper that should certainly eventually appear in Nature Communications -- but perhaps after one round of revision to address the comments above.

Reviewer #2 (Remarks to the Author):

K Singh et. al. study the interacting Hofstadter spectrum of twisted bilayer graphene using the heavy fermion model introduced by Song and Bernevig. The authors analyze the interacting Hofstadter model and claim to provide an analytical understanding of the interacting Hofstadter spectrum. The results make physical sense. Understanding the interacting Hofstadter spectrum is a hard problem, especially in the magic angle regime, and the topological heavy fermion model is certainly a novel attempt.

However, the paper is highly technical, and the results lack novelty worthy of publication in Nature Communication. In my opinion, the paper needs a significant rewrite with a particular emphasis on clarification of the physical results. It might also be worthwhile to discuss any experimental consequences. In addition, I have a few comments and questions about the paper.

1) Can the topological heavy fermion model be extended to non-magic twist angle regimes?

2) As I understand, the authors consider only translational invariant broken symmetry states in their analysis. What about valley density waves, spin density waves, and Kekule states at integer filling factors? Experiments at fractional filling in tBLG show evidence of charge density wave states (see Nature, 600, 439 (2021)). Is there a reason why such states would have higher energy when compared to the translationally invariant states?

While the technical aspects look good, in my opinion, the paper is not clear. If there is a significant rewrite of the paper with a discussion of experimental consequences, then I think the paper should be considered for publication in Nature Communications.

Reviewer #3 (Remarks to the Author):

Recently, the topological heavy fermion model (THFM) of magic angle twisted bilayer graphene has aroused significant research interest. In this manuscript, the authors extended the THFM to include orbital magnetic effects in such a way that the Hofstadter energy spectrum is interpreted as the Landau levels of itinerant Dirac fermions coupled with strongly correlated, localized $px \pm ipy$ like orbital. It is remarkable that exact results for the interacting Hofstadter spectrum (in the presumed valley polarized state) are obtained in the flat band limit. Even away from the flat band limit, the authors managed to numerically calculate the Hofstadter spectrum at unprecedentedly small magnetic fields. The results presented in this work offer an intuitive understanding to the delicate structure and the anomalous low energy mode in the Hofstadter spectrum of magic angle TBG, which would be of great interest to the community of moire superlattices and quantum Hall physics. I would like to recommend this work for publication in Nature Communications. I also have some questions and comments which I hope the authors could consider:

1. The authors assume a valley polarized state at different integer fillings. What if the ground state is an intervalley coherent state? Part of the orbital magnetic effects are to induce orbital Zeeman splittings and to change the population of the Chern band due to Streda formula. In this sense, an intervalley coherent ground state are expected to behave quite differently from a valley polarized one under magnetic field. It would be nice if the authors could comment on the difference.
2. Heterostrains and lattice relaxations are quite common in TBG device. Strain would open up a gap between remote bands and flat bands, and would also strongly break the particle hole symmetry. Could the authors briefly comment on the effects of strain fields on the Hofstadter spectrum?

We appreciate the reviewer for their time and effort dedicated to the assessment of our manuscript, “Topological Heavy Fermion in Magnetic Field”. We are elated to have found that they perceive the manuscript as a value addition to the existing literature. We highly value the detailed and insightful suggestions for improvement made by the reviewer and have tried to incorporate them in the revised draft. Below we list the revisions based on the suggestions by the reviewer and our response to the comments and enquiries. The reviewer’s comments and enquiries are labeled in red and our response to it is labeled in blue. Mathematical objects are labeled in black while vectors are labeled in bold black.

Reviewer Comment 1:

“In fact, while overall I ultimately lean to recommending publication, my main concern stems from the technical level of the work -- it is fairly heavy going even for experts, and the pedagogy, while refreshingly direct and minutely detailed, occasionally can lead to a loss of the "big picture" due to the weighty formalism that is (rightly) introduced to solve the problem. Therefore, while I think that overall the work is very good, my recommendation would be to improve the presentation to add pedagogical material and some more pointers to the uninitiated.”

Authors Response 1:

We thank the reviewer for suggesting an improvement in the presentation of our results to benefit the clarity of disposition. To reduce the technical aspect of the presentation and improve upon the pedagogy, we have substantially revised the layout of the draft. We hope that our efforts to address these concerns have substantially improved the manuscript and made the material accessible to a larger audience suitable for Nature Communications, in accordance with the Reviewer’s observations.

Reviewer Comment 2:

I think it would be very helpful for the authors to give an overview of why just minimal coupling of the f-electrons won't work, to introduce the reader to the key challenge addressed here (maybe a good place would be the third paragraph of the introduction).

Authors Response 2:

We thank the reviewer for the suggestion. In the revised draft, the first section in ‘**Results**’ dedicatedly motivates the key challenge involved in promoting the THFM in the presence of an out-of-plane magnetic field \mathbf{B} . We present the exact results obtained via naive minimal coupling for a simplistic scenario, where we set the bandwidth of the narrow bands ‘ M ’ and spatial width of the localized Wannier states ‘ λ ’ to zero, and show that we do not recover the correct total count of two zero modes per moiré unit cell per valley per spin. For the general case, we refer the reader to **supplementary note 8**, where we show that the naive minimal coupling approach still does not recover the correct total count of two modes within the narrow bands per moiré unit cell per valley per spin.

Reviewer Comment 3:

I just want to confirm that the notation in eq(5) is consistent with Ref[39] -- there the J and U_1 are linked to quartic terms, but I took eq (5) (per the statement "mean field interaction") to just come from a replacement of these with the appropriate quadratic mean-field term -- is this correct?

Authors Response 3:

This is a completely correct observation by the reviewer. Since we incorporate the interactions at a mean field (MF) level, we have replaced all the quartic interaction terms with appropriate quadratic

counterparts. In the revised draft, we provide the MF interaction for the coupled modes in a matrix form (in Eqs.(23),(25) and (27) corresponding to fillings $\nu = 0, -1$ and -2 respectively) with respect to the spinor in Eq.(17). The MF interaction for the decoupled f -modes is simply stated, which is given by the element at index $\alpha, \alpha' = 5$ and $\alpha, \alpha' = 6$ of the above matrices.

Reviewer Comment 4:

Also in the intro, the authors describe the anomalous almost B-independent mode, in terms of the zero-LL of a massless Dirac particle. My understanding is that this is true only for exactly flat bands ($M=0$), and that there would be a weak dependence for non-zero M . Is this correct? I mention this because otherwise I don't see a zero-energy Dirac fermion in the problem -- I thought that for $M \neq 0$ there was only a quadratic band crossing?

Authors Response 4:

For non-zero M , the weakly magnetic field dependent anomalous mode in the spectrum is indeed no longer exactly the same decoupled c -mode provided in Eq.(19) (or supplementary Eq.(259) for valley K'), i.e. the zero-LL of a massless Dirac particle. Using the effective model presented in the revised draft's Eq.(24) with the off-diagonal a^3 / a^{*3} terms dropped and the M term in line '598' included, one can see that the weakly magnetic field dependent anomalous mode is still well separated from the rest of the spectrum above a small B . This indeed is also seen in Fig.(3a) which shows the exact unapproximated THFM Hofstadter spectrum at CNP. Indeed the physical nature of the anomalous mode above a small B stays unaltered even when we tune the bandwidth from 0 to M .

Reviewer Comment 5:

In Fig. 1, 2, I think the spectrum is shown for both valleys, correct? If not, I don't immediately see how one can get a level at both $\pm J/2$ on the basis of Eq. (5). It might be worth flagging in different cases when one is talking about single-valley spectra and where one is discussing the both valleys (and both spins, where relevant)

Authors Response 5:

This is a completely correct observation by the reviewer and we thank them for the suggestion. We consider both the valleys in Fig.(2b) (which was Fig.(1b) in the previous draft) and thus have both the $\pm J/2$ modes. In the previous draft, Fig.(2) presented the spectrum only at valley K' (this figure is not included in the revised draft). We have added the 'spin-valley' sector (wherever missing) to the caption of each figure presenting the Hofstadter spectrum in the revised draft.

Reviewer Comment 5:

I think it would be worth crisply defining that "flat band" is equivalent to " $M=0$ " somewhere and that this is the terminology used throughout. Often people colloquially use flat and almost-flat interchangeably as in the "flat bands of moire graphene" even when the flatness is not exact. My understanding is that the authors are being (appropriately) more precise, but it's good to say this.

Authors Response 5:

We thank the reviewer for the suggestion. Within the **Introduction**, in lines '57-58', we have explicitly stated the flat band condition to be $M = 0$. Also, in the revised draft we always refer to the bands as 'narrow bands' whenever we discuss the case with $M \neq 0$.

Reviewer Comment 6:

What is the meaning of \mathbf{p} in equations (6) and (7) -- it does not appear on the RHS, so it's a bit confusing. I suspect that perhaps by \mathbf{p} here one means the appropriate operator identification $\mathbf{p} \rightarrow -i \hbar \nabla$, but I might have misunderstood this completely. It is not central to the discussion but would be good to clarify.

Authors Response 6:

We thank the reviewer for pointing out the missing definition for \mathbf{p} in the previous draft. As correctly suspected, \mathbf{p} in Eqs.(6) and (7) of the previous draft referred to the momentum operator. We do not have these equations in the revised draft.

Reviewer Comment 7:

When discussing the degeneracy of each LL (line 199) it might be good to remind the reader that $N\phi$ is the total flux through the system in the authors' convention, so that it's the familiar expression for the degeneracy.

Authors Response 7:

We thank the reviewer for the suggestion. We do not have this statement in the revised draft.

Reviewer Comment 8:

From eq(20), is it correct to conclude that the explicit dependence of \mathbf{A} is unimportant to the c - f coupling, but only the implicit structure deduced from the structure of the LL wavefunctions of c - and f - electrons? If so, it's worth saying so explicitly.

Authors Response 8:

As correctly concluded by the reviewer, the contribution of the vector potential \mathbf{A} (in Landau gauge) is insignificant to the c - f coupling. We do not have the Eq.(20) in the revised draft. However, we do provide the key steps in the derivation of finite magnetic field c - f coupling in **Methods A**. In lines '1169-1179', we explain the steps leading to the above conclusion.

Reviewer Comment 9:

The SVD approach is very elegant but could perhaps be summarized more crisply.

Authors Response 9:

We thank the reviewer for the suggestion. In the revised version, we have tried to adequately summarize the singular value decomposition (SVD) approach. The revised version can be found in the section "Analytical results for the non-interacting Hamiltonian at $\mathbf{B} \neq 0$ " of **Results**. The key modifications in its presentation are as follows: We first motivate the SVD approach by alluding to the fact that it guarantees a lower bound on the total number of zero modes for the non-interacting problem. We next explain that these zero modes are linear combinations of different f -modes, which decouple from the c -modes, and how they can be identified using the SVD approach. Finally, we explain how it helps us to recast the non-interacting THFM in terms of the 6×6 operator in Eq.(18).

We appreciate the reviewer for their time and effort dedicated to the assessment of our manuscript, “Topological Heavy Fermion in Magnetic Field”. We are elated to have found that they perceive the manuscript as a valuable addition to the existing literature, and we understand that the technical nature of our first draft has made its significance obscure. Inspired by the comments, inquiries, and constructive criticisms offered by the reviewer, we have endeavored to improve the clarity and increase the impact of the manuscript. In light of this and the positive reviews of the other referees, we would deeply appreciate the reviewer giving our revised manuscript a second look. For their convenience, we have addressed each comment specifically below. The reviewer’s comments and enquiries are labeled in red and our response to it is labeled in blue. Mathematical objects are labeled in black.

Reviewer Comment 1: “However, the paper is highly technical, and the results lack novelty worthy of publication in Nature Communication. In my opinion, the paper needs a significant rewrite with a particular emphasis on clarification of the physical results. It might also be worthwhile to discuss any experimental consequences.”

Authors Response 1:

We thank the reviewer for suggesting an improvement in the presentation of our results to benefit the clarity of disposition. We have substantially revised the layout of the draft to reduce the technical aspect of the presentation and focus more on elucidating the physical results. We hope that our efforts to address these concerns have substantially improved the manuscript and made the material accessible to a larger audience suitable for Nature Communications.

Motivated by the suggestion of the reviewer, we discuss the experimental consequences by stating the Landau level (LL) sequence at each of the discussed integer fillings $\nu = 0, -1$ and -2 . Moreover, as we show in the main text, these results can lucidly be followed through the effective model analysis presented in the revised draft.

- 1) In lines ‘538-596’, we state and explain the expected Landau level (LL) sequence in the flat band limit, $M = 0$, and at filling $\nu = 0$. In lines ‘597-609’ and ‘610-626’, we explain how to account for non-zero M for valley polarized (VP) and Kramers intervalley coherent state (KIVC), respectively. We further explain the consequence of including M upon the expected Landau level (LL) sequence. The LL sequence ‘ $0, \pm 2, \pm 4$ ’ obtained through this analysis at $\nu = 0$ is in agreement with the results reported in the experiment of Ref [1].
- 2) In the lines ‘720-770’ and ‘845-872’ we state and explain the LL sequence at fillings $\nu = -1$ and -2 , respectively for the VP state. In the above-referred lines, we further explain the consequences of flat band limit and/or asymptotically small limit of the magnetic field upon the LL filling sequence.

Reviewer Comment 2: Can the topological heavy fermion model be extended to non-magic twist angle regimes?

Authors Response 2:

We thank the reviewer for the question. While the model can easily incorporate f -mode kinetic energy, corresponding to the broader bands away from the magic angle, this increased kinetic energy will decrease the density of states and weaken the correlated “heavy fermion” physics that the model is designed to reveal. (See [2] for a construction of the Heavy Fermion model over a range of twist angles.) For non-magic angles, correlation effects are weakened by f -mode broadening, and the model will more

closely resemble a Hofstadter tight-binding model on the triangular lattice with no symmetry breaking. Since this regime is not typically desired in most experiments, we restrict our analysis to magic angle parameters where unexpected quantization rules are in need of explanation.

Reviewer Comment 3:As I understand, the authors consider only translational invariant broken symmetry states in their analysis. What about valley density waves, spin density waves, and Kekule states at integer filling factors? Experiments at fractional filling in tBLG show evidence of charge density wave states (see Nature, 600, 439 (2021)). Is there a reason why such states would have higher energy when compared to the translationally invariant states?

Authors Response 2:

We thank the reviewer for the question. In the revised draft, we further extend our analysis to the Kramers intervalley coherent state (analysis can be found in **supplementary note 9**), which multiple studies have confirmed to be the ground state in the strong coupling regime at fillings $\nu = 0$ and -2 . In the absence of perturbations such as heterostrain and lattice relaxation effects, the strong coupling regime is justified and we would argue that the energetics would favor the translational symmetry preserving generalized quantum Hall ferromagnets at integer fillings, as was seen in Ref[3].

The central result of our paper is a Landau level formalism applicable to any ‘heavy-fermion-like model’. As the Reviewer appropriately points out, additional perturbations to the Bistritzer Macdonald (BM) model stabilize other groundstates not considered in our analysis. However, these terms are also treatable in the heavy fermion formalism and the subject of work currently underway. Thus the method developed in the current paper can be applied to compute their spectrum in the magnetic field. The generality of the formalism is a strength of this work which we hope makes it appropriate for Nature Communications.

REFERENCES

- (1) X. Lu et al, Nat 574, 653-657 (2019).
- (2) D. Călugăru et al, arXiv : 2303:03429 (2023).
- (3) Y. H. Kwan et al, Phys. Rev. X 11, 041063 (2021).

We appreciate the reviewer for their time and effort dedicated to the assessment of our manuscript, “Topological Heavy Fermion in Magnetic Field”. We are very pleased to have found that they perceive the manuscript as a valued addition to the existing literature. We graciously welcome the comments and enquiries made by the reviewer and below we list our response to it. The reviewer’s comments and enquiries are labeled in red and our response to it is labeled in blue. We hope that the edits below will satisfy the Reviewer and make our paper ready for publication. The reviewer’s comments and enquiries are labeled in red and our response to it is labeled in blue. Mathematical objects are labeled in black while vectors are labeled in bold black.

Reviewer Comment 1:

The authors assume a valley polarized state at different integer fillings. What if the ground state is an intervalley coherent state? Part of the orbital magnetic effects are to induce orbital Zeeman splittings and to change the population of the Chern band due to Streda formula. In this sense, an intervalley coherent ground state are expected to behave quite differently from a valley polarized one under magnetic field. It would be nice if the authors could comment on the difference.

Authors Response 1:

We thank the reviewer for the question. Motivated by the comment of the reviewer, we extend our analysis to also include the Hofstadter spectrum for Kramers intervalley coherent state (KIVC) states in the revised draft. We focus on the fillings $\nu = 0$ and -2 because multiple studies (including Ref[1]) have confirmed the ground state to be KIVC in the strong coupling regime at these fillings. We wish to point out that in the flat band limit ($M=0$), the Hofstadter spectra are exactly the same for the valley polarized (VP) and KIVC states. This is due to the spin-valley $U(4)$ symmetry. The differences appear away from the flat band limit when M is non-zero. We discuss the corresponding Hofstadter spectra in the **supplementary note 9**. The main differences compared to the Hofstadter spectrum of VP state are as follows:

- (a) $\nu = 0$: The weakly magnetic field \mathbf{B} dependent anomalous mode that decouples from the rest of the spectrum emanates out of the energy $\pm(J^2/4+M^2)^{1/2} = \pm 9.70$ meV for the KIVC state, as can be seen in supplementary Fig.(11b). Contrary to it, the weakly \mathbf{B} dependent anomalous mode for the VP state emanates out of the energy $\pm(J/2-M) = \pm 5.88$ meV, as can be seen in main-text Fig.(3a). Another difference is that apart from the above anomalous mode, the Landau levels (LLs) for KIVC occur in pairs approaching double degeneracy in the asymptotically small limit of \mathbf{B} , which is not the case for VP state at non-zero M . This can be seen by comparing the supplementary Figs.(14b) and (13b).
- (b) $\nu = -2$: The charge -1 excitation (which is the light mass excitation) at $\nu = -2$ occupies the lower energy band shown in supplementary Fig.(12a), which resides in the energy window ‘ $-2W_3-(J^2/4+M^2)^{1/2} = -108.35$ meV to $-5U_1/2-12U_2 = -161.17$ meV’ and is contributed by spin-up sector of mean-field Hamiltonian. The corresponding Hofstadter spectrum is shown in supplementary figure (12b). The weakly \mathbf{B} dependent anomalous mode that decouples from the rest of the spectrum emanates out of the energy $-2W_3-(J^2/4+M^2)^{1/2}$ for the KIVC state. Due to the overlapping magnetic subbands coming from the Landau quantization of the higher energy band (which also is contributed by spin-up sector) that resides in the energy window ‘ $-2W_3+(J^2/4+M^2)^{1/2} = -88.96$ meV to $-3U_1/2-12U_2 = -109.45$ meV’ (see supplementary figure (12a)), the anomalous mode is not visible in the supplementary figure (12b). However, the existence of the anomalous mode can still be verified through the supplementary Fig.(18c), wherein the magnetic subbands

coming from the Landau quantization of the higher energy band do not overlap with it. This is because we set a very small upper cutoff on the LL-index for the spectrum in this figure to be tractable. We moreover argue that even for large LL-index cutoff, performing self-consistent HF calculation beyond the one-shot HF will make the anomalous mode visible. This is because the same procedure at $\mathbf{B} = 0$ separates the two bands further in energy by a gap (see supplementary Figs.(S8a) and (S8d) in Ref-[1]). Contrary to it, the weakly \mathbf{B} field dependent anomalous mode for the VP state emanates out of the energy $-2W_3-J/2+M = -104.54$ meV as can be seen in ‘right panel-b’ of supplementary Fig.(17).

Similar to as seen at $\nu = 0$, apart from the above-discussed anomalous mode, the LLs for KIVC occur in pairs approaching a double degeneracy in the asymptotically small limit of \mathbf{B} , which is not the case for the VP state. This can be seen by comparing the supplementary Fig.(18c) with the ‘right panel-b’ of supplementary Fig.(17).

Reviewer Comment 2:

Heterostrains and lattice relaxations are quite common in TBG device. Strain would open up a gap between remote bands and flat bands, and would also strongly break the particle hole symmetry. Could the authors briefly comment on the effects of strain fields on the Hofstadter spectrum?

Authors Response 2:

As correctly pointed out by the reviewer, heterostrain will split the density of states of the narrow bands, breaking the particle-hole and rotational symmetry while preserving flatness away from the Gamma point. Consequently, the kinetic energy would increase and become comparable/larger to the energy scale of the Coulomb interaction ~ 20 meV [3]. This shifts us from a strong coupling to an intermediate coupling regime and can stabilize proximate nematic and translational symmetry broken candidate ground states [2,3,4,5]. Accommodating the effects of heterostrain would manifest as additional terms in the THFM and will significantly change the nature of the ground states and the Landau level degeneracies. In fact, a comprehensive understanding of this problem is lacking in the literature so far, but can be addressed using the formalism developed here when applied to a strained heavy fermion model, which is the subject of work currently underway.

REFERENCES

- (1) Z.-D. Song and B. A. Bernevig, *Phys. Rev. Lett.* **129**, 047601 (2022).
- (2) J. Kang and O. Vafek, *Phys. Rev. B* **102**, 035161 (2020).
- (3) D. E. Parker et al, *Phys. Rev. Lett.* **127**, 027601 (2021).
- (4) S. Liu et al, *Phys. Rev. Research* **3**, 013033 (2021).
- (5) Y. H. Kwan et al, *Phys. Rev. X* **11**, 041063 (2021).

REVIEWERS' COMMENTS

Reviewer #1 (Remarks to the Author):

The summary of the manuscript and an initial assessment of the results can be found in my previous report. In the updated version the authors have adequately addressed my concerns which mostly centred on pedagogy/presentation/notation and also answered some minor questions of detail. I think the present version is now suitable for publication in Nature Communications.

Reviewer #3 (Remarks to the Author):

The authors have properly addressed all my questions and comments in the first round. Thus I recommend this manuscript for publication in Nature Communications in the present form.